

**Generating a global gridded tillage dataset**

**Authors:** Vera Porwollik [1]; Susanne Rolinski[1]; Jens Heinke[1]; Christoph Müller[1]

[1]Potsdam-Institute for Climate Impact Research, Member of the Leibniz Association, Potsdam, 14412, Germany

*Correspondence to:* Vera Porwollik (vera.porwollik@pik-potsdam.de)

**Abstract.** Tillage is a central element in agricultural soil management and has direct and indirect effects on processes in the biosphere. Effects of agricultural soil management can be assessed by soil, crop, and ecosystem models but global assessments are hampered by lack of information on type and spatial distribution. This study describes the generation of a global classification of tillage practices and the spatially explicit mapping of crop-specific tillage systems for around the year 2005.

Tillage practices differ by the kind of equipment used, soil surface and depth affected, timing, and their purpose within the cropping systems. We classified the broad variety of globally relevant tillage practices into six tillage systems. The identified tillage systems were allocated to crop-specific cropland areas with a resolution of 5 arc-minutes. The allocation rules were based on literature findings and combine area information on crop type, water management regime, field size, water erosion, income, and aridity. We allocated national Conservation

Agriculture areas to grid cells via a probability-based downscaling approach for 54 reporting countries. The dynamic definition of the allocation rules and accounting for national statistics, such as the share of Conservation Agriculture per country, also allows for deriving datasets for future global soil management scenarios. We present the mapping of six tillage systems: no-tillage in the context of Conservation Agriculture (1.1 Mkm²), traditional annual (4.01 Mkm²), traditional rotational (0.65 Mkm²), rotational (0.74 Mkm²), reduced (0.15

Mkm²), and conventional annual tillage (4.65 Mkm²). Further we identified a total area of 4.67 Mkm² ha as potentially suitable area for Conservation Agriculture under assessed current conditions. We elaborate on the results of a sensitivity analysis for our downscale approach as well compare tillage system area results to literature estimates.

The presented tillage dataset and source code are accessible via an open-data repository for modeling

communities interested in the quantitative assessment of biophysical and biogeochemical impacts of land use and soil management (DOIs: 10.5880/PIK.2018.012 and 10.5880/PIK.2018.013 (Porwollik et al., 2018a, b)).



## 1    Introduction to tillage

Global cropland covers an area of about 15 Mkm$^2$ (Ramankutty et al., 2008), which is approximately 13% of
global ice-free land, whereas cropland and associated land management contributes about 4.5% of global
anthropogenic GHG emissions accounting for emissions from rice cultivation, peatland drainage, and N fertilizer
application in the year 2000 (Carlson et al., 2016). Tillage and plowing (further jointly referred to as tillage) are
practiced on most of this cropland (Erb et al., 2016; Pugh et al., 2015). Tillage comprises farm operations usually
practiced for seedbed preparation, weed and pest control, or incorporation of soil amendments. According to
Schmitz et al. (2015) conventional tillage can be distinguished into traditional systems with manual labor and
tools, and mechanized systems. Conventional tillage usually comprises inversion and mixing of the soil layers
with the biophysical of loosening the soil, leading to altered temperature and soil moisture levels in the affected
soil layer (S1 for further terms and definitions used in this study). Current global soil management practices
trend towards a reduction of tillage operations and intensity (Derpsch, 2008; Smith et al., 2008). Reduced
intensity of the tillage operation as either in the case of strip-, mulch-, ridge- and no-tillage is also referred to as
conservation tillage (CTIC, 2018). Reduced tillage practices are especially suitable for agricultural production
(a) of grain crops such as cereals, legumes, and oilseed crops (Giller et al., 2015); (b) on large, mechanized farms
to save labor (Mitchell et al., 2012; Ngwira et al., 2012), fuel (Young and Schillinger, 2012), and machine
wearing (Saharawat et al., 2010); (c) under arid climate conditions, because of its soil moisture preserving effect
(Kassam et al., 2009; Pittelkow et al., 2015); and (d) on soils with high erosion rates (Govaerts et al., 2009;
Schmitz et al., 2015).

Up to now there has been only little effort in the classification and area assessment of tillage systems at the
global scale. The only global statistical data on a kind of tillage system area is provided by the FAO for the
extent of Conservation Agriculture (CA) area (FAO, 2016) at the national scale, which soil management concept
includes minimum soil disturbance (direct seeding techniques), a permanent organic soil cover as mulch or green
manure, and a diversified crop rotation (Kassam et al., 2009). CA covers about 10% of the global cropland area
(FAO, 2016). The top-three adopting countries of CA in terms of area are Argentina, Paraguay, and Uruguay
(73.51%, 66.67%, 46.13% of their arable land respectively) (FAO, 2016). Widest area spread of CA practice is
reported for South America followed by North America (accounting for over 84.6 % of total global CA area),
where it has been originally developed. Adoption of CA is much lower in Europe, Asia, Australia & New
Zealand, and with lowest adoption rate in Africa (1.1%, 2.3%, 11.5%, 0.3% of reported total global CA area
respectively) (Derpsch et al., 2010).

Prestele et al. (2018) mapped reported national values of CA area reported by Kassam et al. (2015) to HYDE
cropland (Klein Goldewijk et al., 2011) for the year 2012. For downscaling national values Prestele et al. (2018)
developed a CA adoption index per grid cell composed by a set of spatial predictors as aridity, field size, soil
erosion, market access, and poverty, based on literature findings resulting in a map at a spatial grid resolution of
5 arc-minutes available to interested users.

Data on tillage practices are available, e.g. for the USA through the reporting of the National Crop Residue
Management    Survey    published    by    Conservation    Technology    Information    Center
(http://www.ctic.purdue.edu/CRM/crm_search/, accessed 08/21/2018). The survey was pursued at national level
until 2004 and continued for a subset of counties for subsequent years reporting on farming area managed under
conventional, reduced, and conservation tillage (with their sub-categories of no-, ridge-, and mulch-tillage). For
Europe, tillage practices have most recently been assessed by the Survey on agricultural production methods



(SAPM) in 2010 based on census and sample survey data and published by EUROSTAT

(http://ec.europa.eu/eurostat/statistics-explained/index.php?title=Glossary:Survey_on_agricultural_production_methods_(SAPM), accessed 08/23/2018). In the EUROSTAT data portal farm type and size, and their corresponding area managed under the tillage categories: conventional, conservation tillage, and zero-tillage (often used as a synonym for no-tillage as referring to direct seeding techniques) are reported. Analyzing tillage practices in the EU-27 for the year 2010, it

has been found that on average the share of conservation and zero-tillage practices increases with the size of the arable land area of a farm holding (EUROSTAT, 2018).

Erb et al. (2016) reviewed data availability of land management practices at the global scale and found that there was no continental or global dataset on area, distribution, and intensity of tillage practices. They report 7.43 Mkm² of cropland comprising annually harvested area to be under high intensity tillage and 4.73 Mkm² of area

under low intensity tillage, which comprises the cropland area of perennial crops, zero-tillage as stated by Derpsch et al. (2010), and young and temporal fallow cropland area as reported by Siebert et al. (2010).

Soil, crop, vegetation, erosion, and Earth system models (in the following jointly referred to as ecosystem models) can be applied to assess the effect of different tillage practices on ecosystem elements fluxes and stocks. Some global carbon studies assess the climate mitigation potential of soils managed with no-tillage compared to

conventional tillage, which was simulated as a temporally limited enhancement of the decomposition factor on the soil carbon pools under cultivated cropland (Levis et al., 2014; Olin et al., 2015; Pugh et al., 2015; Smith et al., 2008). More process-based representations of the tillage effect are applied in models as CSM for DSSAT (White et al., 2010), and CROPGRO-soybean (Andales et al., 2000) having direct and indirect effects on soil, water, crop yield, and emissions. Another field of global studies assessing the tillage effect refers to the analysis

of albedo enhancement perceived in cases of no-tillage in conjunction with associated increased residue levels left on the surface of the field (Hirsch et al., 2017; Lobell et al., 2006). Furthermore, tillage is important in soil erosion assessment studies, often represented within the context of the land management factor amplifying sub-factors as surface cover and surface roughness (Nyakatawa et al., 2007; Panagos et al., 2015).

McDermid et al. (2017) reviewed regional models and ESMs' approaches of representing agricultural

management practices and land use conversion with a focus on climate and land surface interactions, including tillage modifying carbon stocks in the soil as well as biogeophysical surface attributes. They reveal sources of uncertainty due to missing land management data and limited representation of processes in current assessment models. In regard to tillage implementation in ESMs, they elaborate on the findings of Levis et al. (2014) who found decreased soil carbon levels under cropped and cultivated land compared to land without cultivation.

McDermid et al. (2017) state potential overestimation of efficacy of no-tillage practices contributions to mitigate anthropogenic carbon by enhanced carbon stock based on findings of Powlson et al. (2014). Pongratz et al. (2017) also reviewed data availability and process implementations within ESMs for ten land management practices and resumed tillage to be currently underrepresented. They recommend simple and complex methods to model tillage effects on albedo, soil moisture, respiration, and resulting effects on soil carbon stocks and fluxes.

The ecosystem modeling community relies on sometimes nontransparent assumptions on type and spatial distribution of tillage systems, or can assess different tillage impacts just in form of scenarios (Del Grosso et al., 2009; Olin et al., 2015; Pugh et al., 2015).



The objective of this study is to a) increase understanding of the drivers for different tillage practices and their spatial distribution at the global scale b) develop an open source and open data crop-specific tillage system dataset for the parameterization of tillage events in global ecosystem models and assessments. For this we develop a global tillage system classification. Further we aim to formulate a set of rules by analyzing underlying causes and drivers for the occurrence of different tillage systems and make use of available data in order to map them to a global grid of 5 arc-minutes resolution.

## 2    Data and method

### 2.1    Figure 1Tillage system classification

Globally tillage systems differ by the kind of implement used, soil depth and share of soil surface affected, timing, and by their purpose within the relevant cropping systems (Table 1).

Conventional tillage refers to the inversion and mixing of soil layers for seedbed preparation, weed, pest, residue management, and incorporation of soil amendments. In traditional tillage systems soils are usually managed with hand tools, e.g. hoe or cutlass (Schmitz et al., 2015), which is very labor and time intensive. The application of animal-drawn plows or the use of a moldboard plow attached to some motorized vehicle result in increased soil depth and mixing efficiency of the tillage operation compared to traditional tillage implements. For CA we assume the minimal soil disturbance by direct seeding equipment.

We set timing and frequency of soil disturbance by tillage depending on the type of cropping system. For annual crops, tillage is performed annually at the time of establishment or after harvest. When modelling perennial crops, the interval of main tillage events on fields should reflect the length of the entire perceived plantation cycle. During the year for annual and perennial cropland less intense tillage may be necessary for weed management or intended inter-cropping purposes several times. This soil management is locally restricted to the space between the rows of the main crop and could be replaced by herbicide applications. Within CA managed systems disturbance of the soil occurs only at the time of seeding. Weed in CA systems is either managed by sustaining a permanent soil cover of either mulch or cover crops, by diversified rotations, and by application of herbicide so that no further mechanical soil disturbance is necessary during the growing season.

The soil depth affected by the tillage operation is determined by the soil depth to bedrock, the implement used to till the soil, and by the purpose of the tillage event. A moldboard plow usually inverts and mixes the soil layers up to 20-30 cm depth. Pimental and Sparks (2000) state the minimum soil depth for agricultural production to be 15 cm. Whereas Kouwenhoven et al. (2002) state that for burying green manure and annual weed, a minimum tillage depth of 12 cm to be necessary, and suggest 20 cm for the management of perennial weeds. We decided for a minimum depth of mechanized tillage of 20 cm. For traditional tillage with manual labor, tillage is assumed to reach only to a lesser depth, because of limited capacity to penetrate the soil profile (Schmitz et al., 2015). The affected depth by minimum soil disturbance practices under CA is assumed to be as deep as the seed placement requires, which is stated as approximately 5 cm by White et al. (2010) for no-tillage systems.

Conventional tillage both in mechanized and traditional farming systems leaves a low portion of residues covering the soil surface after seeding - usually less than 15% (CTIC, 2018; White et al., 2010). Reduced tillage may leave 15-30% whereas in CA systems minimum soil surface covered by organic mulch is defined as at least 30% after planting (CTIC, 2018).



Tillage mechanically loosens the soil by decreasing the bulk density. The mixing efficiency of tillage describes the degree of homogeneity achieved when burying crop residues and redistributing soil particles in the affected soil horizon. Soil bulk density and pore space determine the levels of surface contact between seeds and soil particles, root growth, and water infiltration. Soil characteristics as moisture, temperature, are altered by the

mixing effect of tillage. The type of soil, its moisture content, and the speed of the tillage practice are further determining factors for the mixing efficiency of tillage (White et al., 2010) under field conditions. Too intensively or inappropriately tilled soils over a longer time period exhibit the destruction of soil aggregates by increasing bulk density leading to compaction or crusting (White et al., 2010). The mixing efficiency can be modelled as a factor modifying the homogeneity level of soil components and associated characteristics.

In mechanized conventional and traditional tillage systems, the implement is usually applied on the entire soil surface to be effective. In contrast to that, no-tillage under CA maximal may affect 20-25% of the soil surface during the direct seeding procedure (Kassam et al., 2009; White et al., 2010). On the field reduced tillage as partial disturbance of the soil surface in case of strip-, mulch- or ridge tillage can be achieved by applying either an inverting implement to a lesser soil depth or lower share of soil surface affected, by using less soil disturbing

harrows or disks, or by less field passes. Reduced tillage practice could be simulated as with lower soil disturbance frequency, depth, mixing efficiency, or higher residue share left on the soil surface ranging between values of conventional and no-tillage. Based on the literature findings mentioned above we consider six different tillage systems, namely no-tillage in the context of Conservation Agriculture, conventional, rotational, traditional annual, traditional rotational, and reduced tillage (Table 1).

165         (Table 1)

## 2.2    Datasets used for mapping tillage systems to the grid

For mapping tillage systems, spatial indicators on the basis of several environmental and socio-economic datasets are applied (Table 2). The basic data layer to this mapping study is the cropland dataset by the spatial production allocation model further referred to as SPAM2005 by International Food Policy Research Institute

and International Institute for Applied Systems Analysis (IFPRI/IIASA, 2017b). It reports physical cropland area for 42 crop types (Table S2 for a list of crop types), for the year 2005. The spatial resolution of the dataset is 5 arc-minutes. The SPAM2005 dataset comprises four technology levels of crop production, distinguishing high input irrigated from purely rainfed areas with further distinction of rainfed areas into high input, low input, and subsistence production (You et al., 2014). Adding up the reported cropland area of SPAM2005 for 42 crop types

results in a total sum of 11.31 Mkm². The cropland by IFPRI/IIASA (2017b) comes along with a grid cell allocation key to country (IFPRI/IIASA, 2017a), which has been used in this study for any grid cell aggregation to country scale.
Sub-national aggregations of grid cells to state or province level were done with the Global Administrative Areas data base (Global Administrative Areas, 2015).

The dataset on soil depth to bedrock (Hengl et al., 2014) has been retrieved from SoilGrids, which is a soil information system reporting spatial predictors of soil classes and soil properties at several depths. It has been derived on the basis of the United States Department of Agriculture (USDA) soil taxonomy classes, World Reference Base soil groups, regional and national compilations of soil profiles, several remote sensing, and land cover products using multiple linear regressions. The dataset reports on the absolute depth to bedrock (cm) per

grid cell at 5 arc-minutes resolution.



The global gridded field size dataset by Fritz et al. (2015) reports four field size classes as "very small" (smaller than 0.5 ha), "small" (0.5 to 2 ha), "medium" (2 to 100 ha), and "large" (larger than 100 ha) (Herrero et al., 2017) for the year 2005 at 0.5 arc-minutes resolution.

The Global Land Degradation Information System (GLADIS) (Nachtergaele et al., 2011) reports land degradation types and their spatial extent around the year 2000. From this database the global gridded water erosion data has been selected. The water erosion data reports sediment erosion load (t ha$^{-1}$ year$^{-1}$) per 5 arc-minutes grid cell which the authors derived by applying the Wischmeier equation (Wischmeier and Smith, 1978). Values of the data range from 0 to 12,110 t ha$^{-1}$ year$^{-1}$ with highest water erosion levels occurring in mountainous areas.

The aridity index dataset was retrieved from the Food and Agriculture Organization Statistics (FAO, 2015). The aridity index was calculated as the average yearly precipitation divided by the average yearly potential evapotranspiration (PET), based on Climate Research Unit (CRU) CL 2.0 climate data averaged for the years from 1961 to 1990 applying the Penman-Monteith method. The aridity index dataset has a 10 arc-minutes resolution. It reports values per grid cell ranging from 0 to 10.48, where values smaller than 0.05 are regarded as
"hyper arid", 0.05-0.2 as "arid", 0.2-0.5 as "semi-arid", 0.5-0.65 as "dry humid", and values larger 0.65 as "humid".

(Table 2)

The online data base AQUASTAT reports annually the spread of Conservation Agriculture (CA) practices at the national scale (FAO, 2016). From this data source, national CA area values were retrieved for all 54 countries
that reported any CA with the total area sum of 1.1 Mkm². Not all of these countries reported values for the year 2005, so that values closest to 2005 were selected from the available set, giving preference to data availability over matching the year 2005.

The average farm size per country dataset (n=133) (Lowder et al., 2014) is based on FAO farm size time series data. National average farm size was largest in land-rich countries, with the top-three countries being Australia
(3243.2 ha), Argentina (582.4 ha), and Uruguay (287.4 ha) (Lowder et al., 2014). The authors found average farm size to increase with elevated income level of a country.

Further we retrieved the income level per country by World Bank (2017) for the year 2005. The data refers to four categories of countries gross national income (GNI capita$^{-1}$ year$^{-1}$), as "Low income" (less than 875 US $), "Lower middle income" (876-3,465 US $), "Upper middle income" (3,466-10,725 US $), and "High income"
(more than 10,725 US $).

### 2.3   Processing of input data and mapping rules

For calculation purposes, all gridded datasets mentioned above were harmonized in terms of extent, resolution, and origin. The spatial extent of the target dataset comprises all cropland cells reported by SPAM2005 (IFPRI/IIASA, 2017b). Targeted resolution is 5 arc-minutes, which partially required resampling and (dis-)
)aggregation of the datasets using the R (R Development Core Team, 2013) packages 'raster' (Hijmans and van Etten, 2012), 'fields' (Nychka et al., 2016), and 'ncdf4' (Pierce, 2015) (also see accompanying R-code).

We developed several mapping rules have been in order to allocate the derived tillage system mentioned above to the grid scale, employing a decision tree as shown in Fig. 1. The decision tree approach has also been applied in other spatial mapping exercises, e.g. in Verburg et al. (2002) and Waha et al. (2012). Hierarchical



classification procedures based on expert-rules can be used to distribute data of a larger spatial units to the grid cell level (Dixon et al., 2001; Siebert et al., 2015; van Asselen and Verburg, 2012; van de Steeg, 2010).

As a first step, the SPAM2005 cropland dataset is masked for grid cells reporting cropland but soil depth to bedrock of less than the required 15 cm for agricultural production according to Pimental and Sparks (2000) (Fig. 1). The entire cropland of these shallower grid cells is allocated to the reduced tillage system, where tillage

practices as ridging or raised beds may be practiced by the farmer, because of physical hindrance for inverting tillage practices at increased depth.

The remaining cropland is separated into annual and perennial cropland following Erb et al. (2016)'s findings, differing between plant type associated tillage by intensity (Table S2 for crop type classification).

Annual and perennial tillage systems are further distinguished by the level of mechanization and commercial

orientation of the crop production unit. We follow the definition used in Lowder et al. (2016) for smallholder farming, if cultivation area is smaller than 2 ha. Levin (2006) found that field size and farm size are positively related and according to Fritz et al. (2015), field size can be regarded as a proxy for agricultural mechanization and human development. Based on these findings, we apply the field size dataset as a proxy for farm size and mechanization. We categorize cropland per grid cell reporting field size equal or larger than 2 ha as 'large' scale

with access to mechanization and field size smaller than 2 ha as 'small' scale farming with rather manual labor. Field size data is not available for all grid cells where SPAM2005 reported cropland. Consequently we interpolated for missing field size grid cell values, using the mean of surrounding grid cell values. The spatial distance to the Hawaiian Islands was too far for this operation, so there field size has been set to value of 2 ha, assuming a land restriction to field size due to the island's geographic pattern and in absence of any alternative

information.

We further assume that animal draught power and mechanized soil management practices on a farm occur as a function of income, indicating the financial capital a farmer might have access to. Therefore, we additionally apply the national average income level dataset to differentiate between small field sizes in higher income countries, where access to financial capital for investment into farm equipment is perceived easier than for

farmers with small field sizes in lower income countries. In order to do so, we summarized countries considered "low" and "lower-middle income" as 'low income', and those countries formerly considered "upper middle" and "high income" as 'high income'. In grid cells reporting newly derived small field size and low income, we then allocated perennial cropland to traditional rotational tillage and annual cropland to traditional annual tillage. In high income countries or in a grid cell reporting field size larger than 2 ha situated in low income countries,

perennial cropland was assigned to rotational tillage and annuals' cropland to conventional tillage assuming a rather commercially oriented farming system with access to market, financial capital, and therefore mechanized soil management equipment (Fig. 1).

As a further step, we distinguished arable production per water management regime following the finding of Kassam et al. (2009) who state, that much of the CA development to date has been associated with rainfed arable

crops. We assumed that soil of irrigated crops is more regularly exposed to some level of mechanical soil surface alteration by farming practices, because efficient and equal distribution of water requires some leveling off of the field to flatten the surface in order to distribute irrigation water most efficient and homogeneous over the field. We allocated all irrigated annual cropland area either to annual traditional or conventional tillage area depending on field size and income level (Fig. 1).



Cropland areas of 22 annually planted rainfed crop types were considered as suitable for CA practice. All annual
      rainfed tubers or rice croplands are excluded from the CA-suitable area following Pittelkow et al. (2015), who
      reported larger yield penalties for these crop types when applying no-tillage practices. Rice is often produced as
      paddy rice, requiring puddling, which is a practice modifying the soil aggregates a lot in order to facilitate the
      steady flooded condition, e.g. to suppress weed growth. We applied a downscale algorithm on the CA-suitable

cropland area (see Fig. 1 box "Downscaling"; see following section for more details), so that part of this
      cropland was assigned either to CA area or checked for soil depth to bedrock again. In case of not being included
      in the CA area and soil depth to bedrock lower than 20 cm, the cropland was assigned to reduced tillage,
      assuming less depth, frequency, mixing efficiency or alternative cultivation practices or in case of enough soil
      depth it was mapped to the conventional annual tillage system.


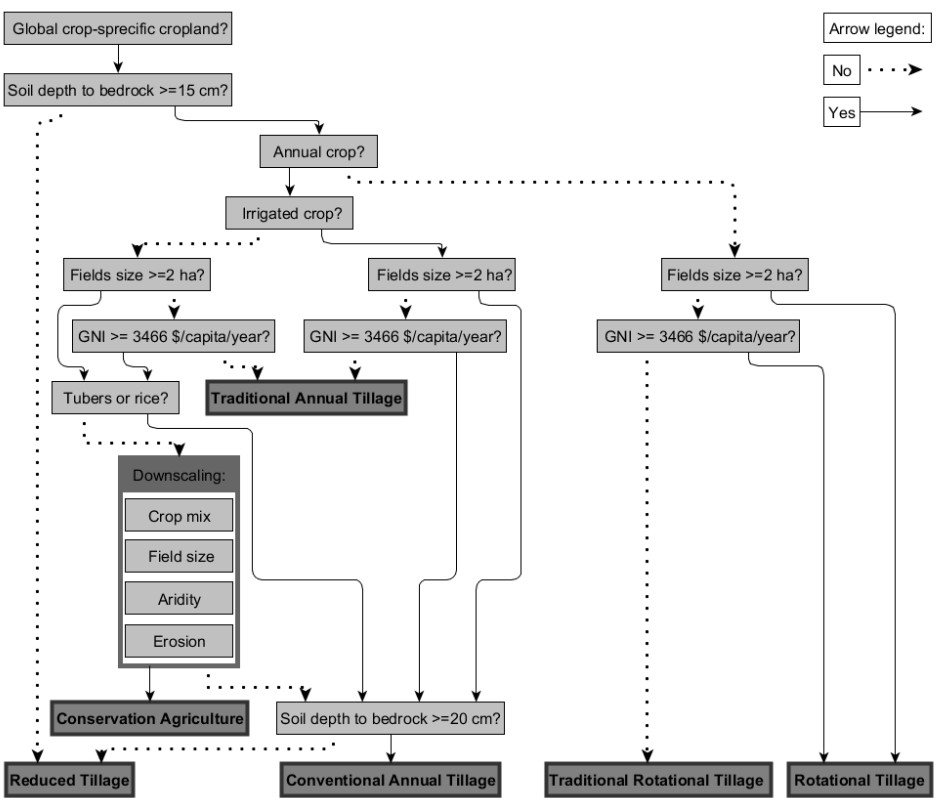

**Figure 1** Decision tree for allocating cropland (ha) to six derived tillage systems. The data processing and
mapping was pursued as depicted from top to bottom of the diagram. Each box represents a check on a grid cell
whether reporting values from the different data layers meet the derived thresholds or specific cropland features.

The arrows with solid lines indicate a 'yes' and arrows with dotted lines a 'no' in the allocation procedure of
      crop-specific area to tillage systems. The box indicating the 'Downscaling' represents our probability and
      suitability indicators applied to downscale national CA area (ha) to a heterogeneous pattern at sub-national scale
      per grid cell. Boxes with darker grey background shading and thicker frames show the derived types of tillage
      systems. (Abbreviation of Gross National Income as: GNI)



### 2.4 Downscaling reported national CA area to the grid cell

#### 2.4.1 Mapping rules for downscaling CA

We selected the following annual crop types reported by SPAM2005 as suitable for CA in this study: barley, beans, chick peas, cotton, cowpea, groundnut, lentil, maize, other cereals, other pulses (e.g. broad beans, vetches), pearl millet, pigeon peas, rapeseed, rest (e.g. spices, other sugar crops), sesame seed, small millet, sorghum, soybean, sunflower, tobacco, vegetables (e.g. cabbages and other brassicas), and wheat (see Table S2) following Giller et al. (2015) findings on CA-suitability for (dryland) grain crop types. Cropland considered to be suitable for CA is derived from the rainfed cropland area of these 22 annual crops (Table S2) in grid cells reporting dominant field size as 'large' and all field sizes in high income countries. The resulting CA-suitable area amounts to 4.65 Mkm². From that CA-suitable area data layer we computed the input variable "crop mix" as the ratio of the sum of 22 CA-suitable crop types' areas over the sum of total cropland area per grid cell.

As stated by Powlson et al. (2014) for the Americas and Australia, by Rosegrant et al. (2014) in general on no-tillage, by Scopel et al. (2013) for Brazil on CA, and by Ward et al. (2018) on CA, largest adoption rates of minimum soil disturbance management principles can be found on medium to large farms. There is few adoption of CA or no-tillage among small-scale farms, with the exception of Brazil (Rosegrant et al., 2014), where adoption of CA is promoted through policies and technological investments.

We developed a linear regression with the 'stats' package of R (R Development Core Team, 2013), applying the linear correlation model ('lm function') to assess the statistical relation between national average farm size (Lowder et al., 2014) and percentage share of CA area (FAO, 2016) on arable land in 2005. The functional relation exhibits an increase in the national share of CA on arable land with an increase in average farm size over the sample (Fig. S3).

Based on the literature findings and regression results, we assumed that no-tillage in the context of CA was highly probable for cropland in grid cells with large fields, here serving as a spatial proxy for large farm size and mechanization.

Furthermore, we considered no-tillage as suitable for arable production under arid conditions (Kassam et al., 2009; Pittelkow et al., 2015), because of less aeration, and more stable pores and soil aggregates compared to soils managed with conventional tillage. In CA systems, the evapotranspiration is additionally reduced by a continuous biomass cover of at least 30% of the soil surface, which promotes yield stability in drought prone production environments.

As a last allocation criterion, CA was regarded as suitable for crop production in areas with elevated erosion levels. Basso et al. (2006) state, that farmers may make use of the green or residue cover to protect the soil surface during high intensity rainfall events. Here the corresponding mapping approach was to assume increased probability of CA practices in cells which report water erosion values exceeding 12 t ha[-1] year[-1] as the upper bound of the soil loss tolerance value (T-values) defined by the USDA (Montgomery, 2007). This assumption also is in line with the finding of Kassam et al. (2009) stating that wind and water erosion are major drivers of CA adoption in Canada, Brazil, and the USA. According to Schmitz et al. (2015) and Govaerts et al. (2009), also Asian and African agricultural producers could benefit from the positive effects of CA in erosion prone areas.



### 2.4.2 Logit model for downscaling national CA

We used water erosion, aridity, field size, and crop mix data per grid cell as spatial predictors determining the distribution of national reported CA area within a country (Fig. S4.1-4). We developed a logit model to

transform and combine values of these four spatial predictor variables into probability values, indicating the likelihood of CA area occurrence per grid cell. We chose the logit model because different ranges of the spatial predictor datasets are made comparable at equal weights without losing much detail. With the help of the logistic regression we deduce the probability of a grid cell to contain CA area as a probability value between 0 and 1.

We applied the spatial predictor crop mix assuming an increasing probability for CA area occurrence in grid

cells, with increasing cultivated area share of CA-suitable crops types. This was based on the assumptions that cropland within a grid cell belongs to one management regime, under which rotations with CA-suitable crops are practiced, and a similar set of soil working equipment is employed. These assumptions also take into account peer group influence and knowledge spillover effects from early adopters of a new technology (here CA practice) towards their neighbors (Case, 1992; Maertens and Barrett, 2013).

Regarding the statistical relation between farm size and CA adoption, we assume that the larger the field size, the higher the CA probability especially for field sizes equal or larger 2 ha depending on the income level of a country, taking 2 ha as the midpoint of the transformed field size logit curve.

We set missing erosion values in grid cells reporting CA-suitable cropland area to the neutral value of 12 t ha⁻¹ year⁻¹, since it depends on very small-scale conditions, e.g. slope. When transforming the water erosion values to

logit, we also set 12 t ha⁻¹ year⁻¹ as the midpoint value of the function.

The midpoint of aridity's logit regression curve is chosen at 0.65 resulting in higher probabilities of CA area occurrence for grid cells reporting arid (values smaller than 0.65) than humid (values larger than 0.65) growing conditions. We interpolated missing aridity dataset where SPAM2005 reports cropland, except for one island grid cell value near Madagascar, which we set to the logit-neutral value of 0.65, because we assume very special

climatic conditions there.

We tested for (Pearson) correlation among the four spatial predictor variables with the R 'base' package (R Development Core Team, 2013), in order to prevent autocorrelation effects (Table 3).

(Table 3)

Generally correlation coefficients (r) among the datasets are low and mostly negative, except for field size and

crop mix.

Those four cropping system indicators are used as explanatory variables in the regression to get the probability of cropland in a grid cell to be CA area as a value between 0 and 1. The probability of CA in a grid cell is derived via the following Eq. (1):

$$CA_{Grid\ cell} = \frac{1}{1+\exp(-\sum_{i=1}^{4} k_i\ (vx_i - xmid_i))} \tag{1}$$

Where, $i$ represents the input datasets of water erosion, aridity, crop mix, and field size (proxy for farm size), $k_i$ refers to the slope value, $xmid_i$ to the central points of each of the logit curves, and $vx_i$ to grid cell values of the referring input dataset.



A sensitivity analysis has been conducted to assess the explanatory power of each of the four input variables (Fig. S5). First step was to vary our chosen reference slope ($k_i$) of each of the input dataset values by factors of 2

and 0.5 (+100%, -50%), as a next step each of the variables is dropped, and finally each of the variables is used as the only variable in the logit model.

### 2.4.3   Mapping CA area per country

Our downscaling of total national CA area values comprises subsetting all grid cells with CA-suitable area per CA area reporting country (FAO, 2016) and then sorting these grid cells per decreasing CA probability values

derived with the logit equation. As a next step, we select grid cells with the top most logit model results while adding up their CA-suitable cropland until the reported national CA area threshold is reached. We received a heterogeneous pattern of allocated CA area at 5 arc-minutes resolution grid within a CA reporting country, according to the likelihood of CA area occurrence based on our logit results and on our statistical data and literature findings.

### 2.4.4   Area potentially suitable for CA

Similar to the 'bottom-up scenario' of Prestele et al. (2018), we deduce potentially CA-suitable area, specifying the socio-economic and biophysical extent of possible CA adoption with respect to crop mix, field size, aridity, and erosion analyzed within this study. We add the subset of 22 annual rainfed crop-specific areas under reduced tillage in grid cells reporting soil depth to bedrock lower than 15 cm, to the CA-suitable area generated.

## 3   Tillage systems per grid cell

### 3.1   Conservation Agriculture area

### 3.1.1   The results of the logit model

We deduced the likelihood of CA area in a grid cell via the logit model approach according to the indicators crop mix, field size, water erosion, and aridity (Fig. 2). The geographical pattern of the logit results (further referred

as ref-logit) exhibits higher probabilities for cropland in grid cells outside the tropical climate zone and in rather continental regions. Probability of CA is higher for cropland in grid cells reporting large field sizes which are mostly found in developed and land-rich countries, i.e. in the USA, Australia, and large parts of Europe. Grid cells in the tropics receive rather low logit results due to their humid conditions, smaller field sizes, lower income levels, and crop types cultivated. In India, China, and Pakistan the majority of cropland was exclude

from CA-suitability.

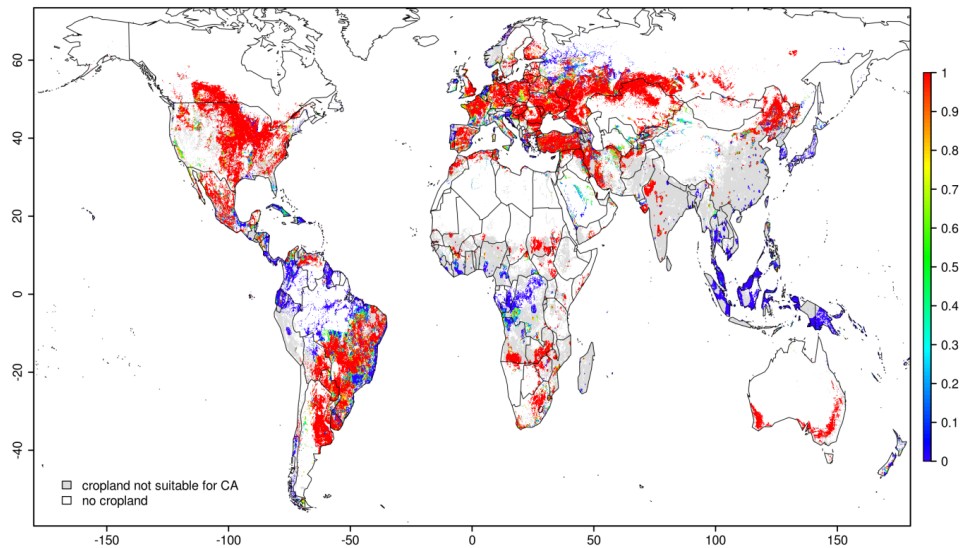

**Figure 2** Probabilities of Conservation Agriculture area per grid cell with high values as red and low ones in blue colors (white color indicates the absence of cropland, and grey the cropland reported by SPAM2005 which is excluded from the area considered suitable for CA due to soil depth, crop type, irrigation, field size, or income level).

### 3.1.2    Results of the sensitivity analysis of the logit model

The sensitivity analysis of our logit model shows mixed responses to our perturbations of slope or variable combination in the logit model (Table 4, Fig. S5). Rank correlation (r) to the ref-logit is much lower when taking one variable only compared to each of the other drop-variable settings or slope modifications. Regarding modifications of the slope parameters of the input variables, we calculated the lowest rank correlation coefficient for increasing the slope of aridity by +100 % and for decreasing the slope of crop mix by -50 % compared to changing the slopes of the other three variables respectively. Erosion has lowest explanatory power as can be interpreted from the very high correlation coefficient to ref-logit when dropping it - but even negative correlation when taking it into the logit equation only.

Our finding is in line with the findings of the sensitivity tests performed by Prestele et al. (2018) who find erosion as the variable with the smallest explanatory power as well.

Crop mix has the largest explanatory power in the logit equation as shown by the lowest correlation coefficient value when dropping it but highest when taking that variable only (Table 4). We additionally report on the sensitivity results for the 54 CA reporting countries, where the effects of slope and variable perturbation show very different patterns per country (Table S6). However, as national CA areas are allocated within individual countries, the sensitivity of ranking within countries is of greater importance than the global rank correlation.

(Table 4)



### 3.1.3 Downscaled CA area

Total downscaled CA area (110,190,763 ha, Fig. 3) is slightly lower than FAO reported total CA area of for these countries (110,289,988 ha). This difference occurs because of our algorithm, which assigned the entire CA-suitable cropland area per grid cell to CA, taking the cropland of the following grid cell in or out of consideration striving for least deviation from the threshold per country (Table S7 for comparison of reported and downscaled country values).


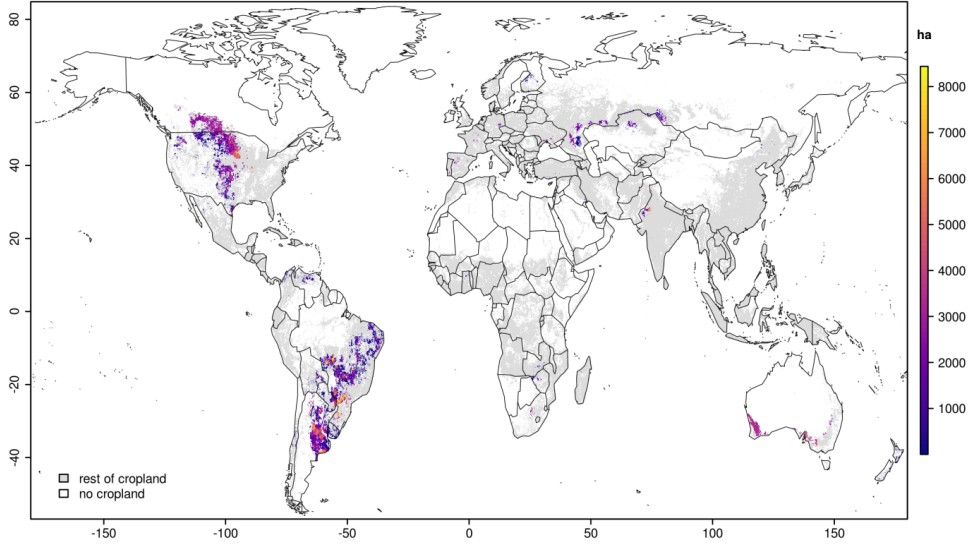

**Figure 3** Downscaled Conservation Agriculture area (ha) (colored) on total cropland (grey) per grid cell for 54 reporting countries around the year 2005.

Aggregated crop-specific areas reveal that most downscaled CA area was allocated to CA-suitable area cultivated with soybean, followed by wheat, and then maize (Table 5). These three crops are among the most important produced, traded, and consumed agricultural goods, making their production highly competitive and therefore the incentive to reduce operational costs (e.g. regarding tillage operation) is high. Another reason for soybean and maize being among the crops mostly produced under CA, may be the usage of high yielding, or

genetically modified crops, coming along with improved pesticide resistances, which make them more suitable for possible herbicide applications (Giller et al., 2015) replacing tillage operations on-field. In Argentina, soybeans are found to be the most common plant cultivated under CA with usually lower residue coverage than required for being a CA system (Pac, 2018). Subsistence farming crops, e.g. peas and millet, were contributing only few cropland to the downscaled CA area (Table 5), because they are more drought resistant (Jodha, 1977),

and of rather regional importance in terms of food security while being traded less on the international markets (Andrews and Kumar, 1992).

(Table 5)



### 3.1.4 Crop specific area potentially suitable for CA

We deduced the total global CA-suitable cropland area of 4.65 Mkm² (see above). Additionally, we identified

0.02 Mkm² of 22 rainfed annual crop types' areas on large fields or in high income countries from the reduced tillage system area, which potentially could be converted to CA area as well. We calculated a total potentially CA-suitable area of 4.66 Mkm², where perceived driving forces, e.g. CA adoption supporting agricultural policies, targeted mechanization efforts, and knowledge dissemination approaches could lead to an area expansion of CA practices.


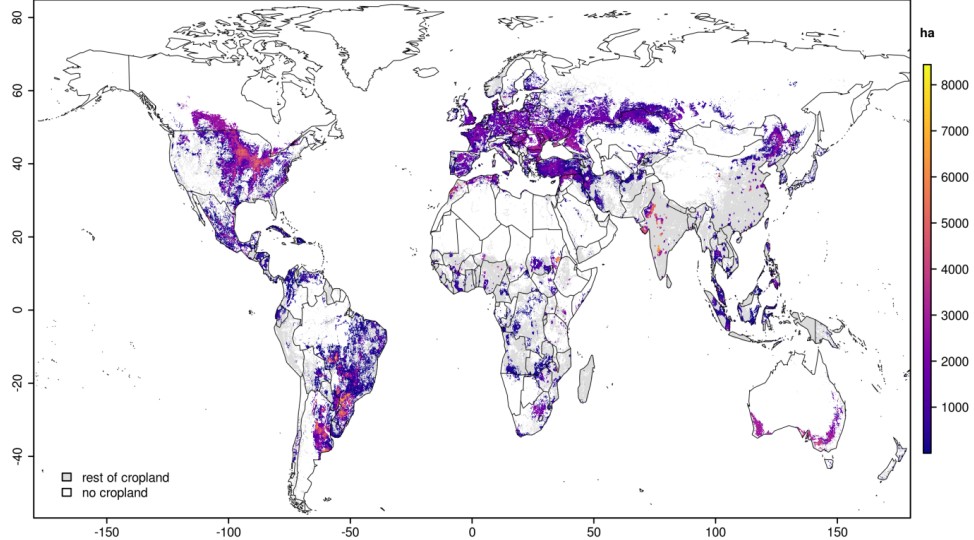

**Figure 4** Area potentially suitable for Conservation Agriculture

### 3.2 Globally total areas and regional pattern of tillage systems

We allocated global cropland of SPAM2005 to the six tillage systems at a spatial resolution of 5 arc-minutes

according to a set of rules. In terms of areas conventional and traditional annual tillage globally constitute the most widespread tillage practices (Table 6). Both systems are applied for annual crops, which globally occupy the largest cropland fraction, are traded, and consumed most. Large parts of the cropland under traditional annual tillage (Fig. 7) for rainfed and irrigated annuals is located in South East Asia, with especially high cropland area shares in India followed by Sub-Saharan Africa, and then South America (Table S9 for aggregated tillage system areas (ha) to country scale). Conservation Agriculture constitutes the third largest tillage system area globally.

Rotational tillage is on the fourth followed by traditional rotational tillage on the fifth position in the ranking of tillage system areas (Fig. 5 and 6). Most traditional rotational tillage system area can be found across the tropical region of South-Eastern Asia and West Africa (Fig. 6). Reduced tillage has the smallest area extent (Table 6) whereas we find most referring cropland in a narrow band between 10° and 20° Northern latitude (Fig. 9). It is

spread in Mexico, African countries Southern to the Sahel zone but mostly found on cropland in India (Table S8 for further metrics across tillage system areas; Table S9).





(Table 6)

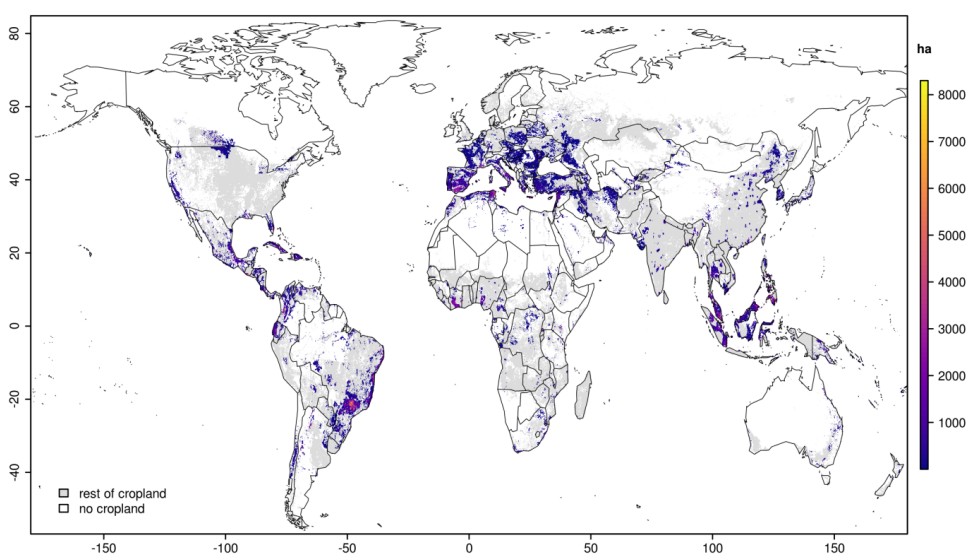

**Figure 5** Rotational tillage area on cropland area of 13 perennial crop types in grid cells with dominating field sizes of minimum 2 ha or larger in low income or all field sizes in high income countries.

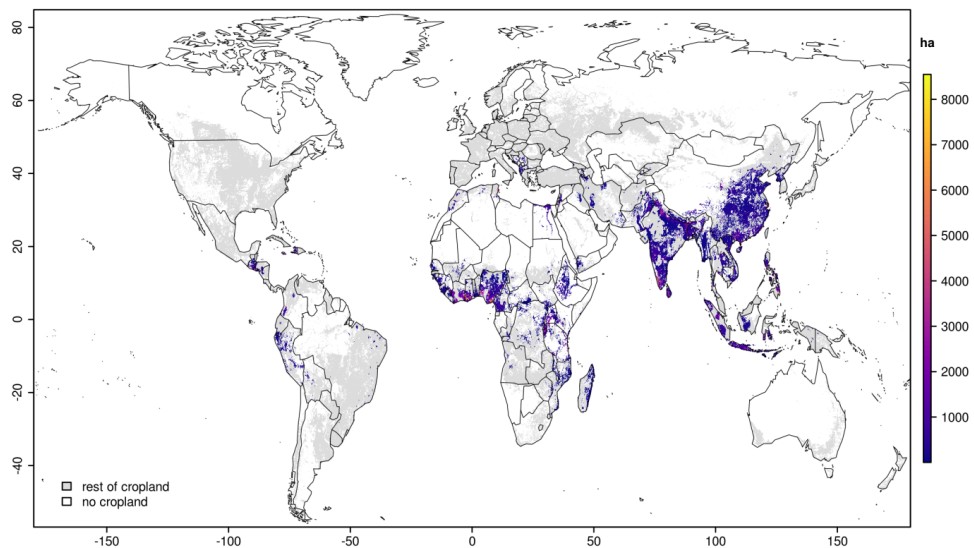

**Figure 6** Traditional rotational tillage area as cropland of 13 perennial crop types in grid cells characterized by field sizes smaller than 2 ha in countries considered as low income in this study.

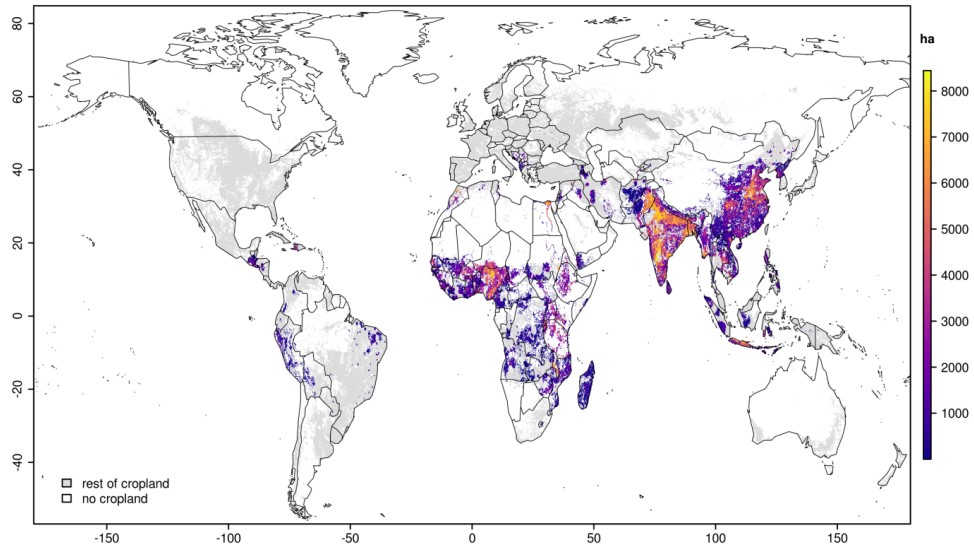


**Figure 7** Traditional annual tillage area as sums over 29 crop types' areas in grid cell reporting dominant field size smaller than 2 ha and in countries classified as low income in this study.

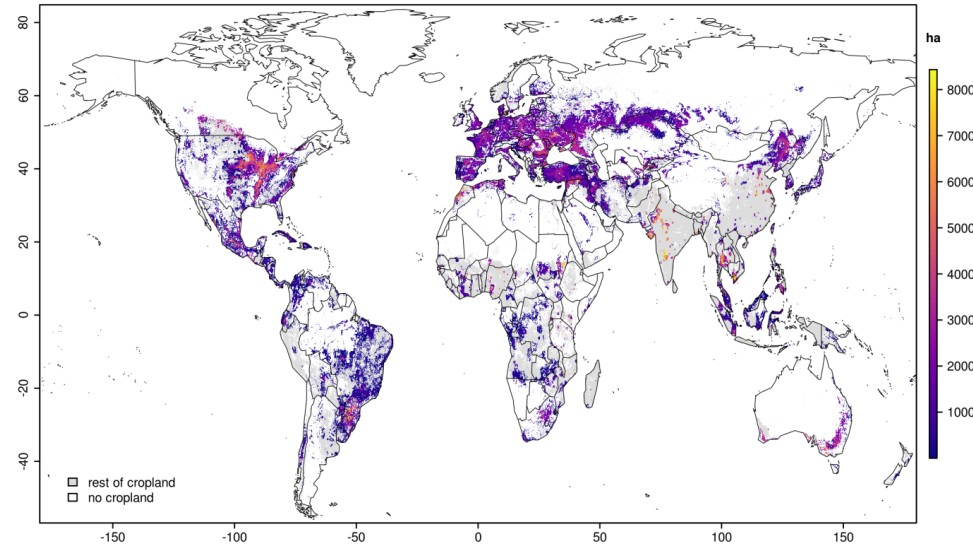

**Figure 8** Conventional annual tillage area, which has been allocated to the majority of global cropland area.



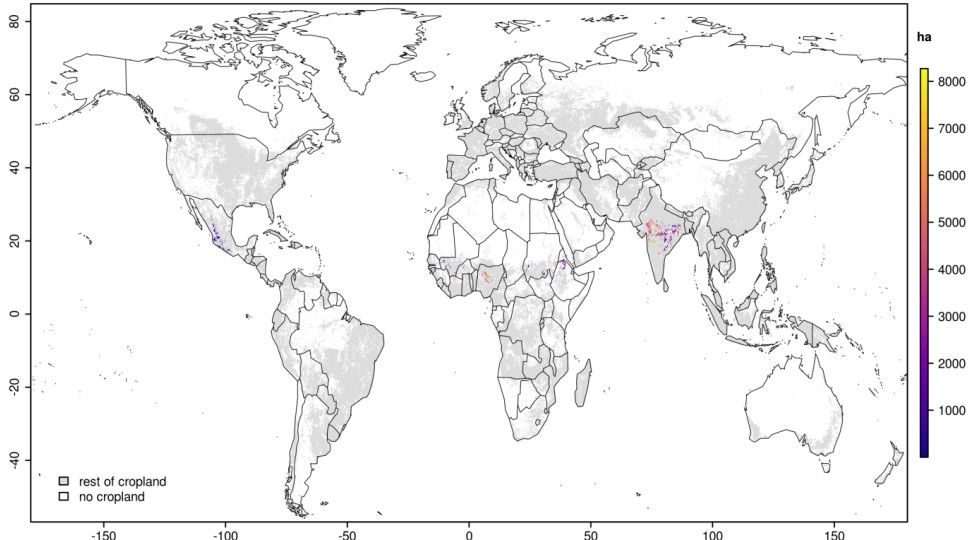


**Figure 9** Reduced tillage area mapped to grid cells reporting soil depth to bedrock shallower than 20 cm, so unsuitable for deep mechanized tillage.

## 4    Discussion

### 4.1    Comparison of results to other studies

In the absence of alternative tillage datasets for validation at the global scale we here want to discuss the way our tillage system area results relate to other studies' findings.

We compare the spatial pattern of our added traditional tillage system area to the one reported by the cropland subsets of SPAM2005 for low input and subsistence farming. According to You et al. (2014), both are production levels characterized by a low level of mechanization or rather manual labor and low input. The sum

of our traditional tillage systems' (rotational and annual) areas (4.63Mkm²) is slightly higher than the sum of SPAM2005 subsistence and low input technological level cropland (4.55 Mkm²). We deduced more traditional tillage system area in South-East Asia, Sub-Saharan Africa, and Peru than SPAM2005 reported under low and subsistence farming (see difference map in Fig. S10). Further comparison reveals a moderately lower amount of area under traditional tillage in our dataset for Europe, the Near East, South America, and Australia, i.e. in

countries which are regarded as emerging or developed economies. The spatial difference may be due to the fact that SPAM2005 is a product of a sub-cell cross-entropy optimization approach to distribute cropland of the same crop species into several production levels per grid cell. Contrary to this, we used the field size and gross-national income as spatial indicators for un-mechanized tillage systems by masking out cropland either per entire grid cell or country-wise according to our derived thresholds. We calculated the spatial correlation via a

regression of the added area values of our traditional tillage system and of the sum of low input and subsistence production level cropland reported by SPAM2005. We found a regression factor ($r^2$) of 0.54 (p < 0.001, slope of 1.139) among both grid cell specific area values.



Our estimate of traditional tillage system area is lower than the finding by Lowder et al. (2016), stating 5.87 Mkm² to be under management of farms smaller than 2 ha size (~12 % of their cropland assumption). Deviations
between the estimates might evolve from our chosen threshold of 2 ha on the field size dataset to distinguish small from large field sizes.

In order to compare our results to the findings of Erb et al. (2016), we added up our reduced, both rotational tillage system areas, and the downscaled CA area to represent the 'low intensity' tillage area, and conventional and traditional annual tillage are summed up to the 'high intensity' tillage area. Since the description of what is
included in their 'low intensity' area is inconsistent within their main text and their supplements, we state two different estimates of our results - both exhibiting different absolute values and shares compared to the findings of Erb et al. (2016) (Table 7).

(Table 7)

Prestele et al. (2018) analyzed CA area time series data by FAOSTAT and have found an increasing trend of CA
adoption within countries and to more countries since the 1970s. The trend is likely going to continue as farm holdings increase in size while decreasing in number in upper middle and high income countries (Lowder et al., 2016). At the same time, the adoption rate of CA in smallholder farming systems in low income countries (e.g. in Sub-Saharan Africa) may persist low in, where average farm size reveals a decreasing trend (Jones, 2017). Adoption of CA practices by smallholder farmers is hampered by competition for residue use (Scopel et al.,
2013), missing knowledge and restricted access to inputs and financial capital (Kassam et al., 2009) making them more risk averse towards adoption of new technology than large scale farmers (Schmitz et al, 2015).

We additionally pursued a provincial and state level comparison between our downscaled CA area to reported no-tillage area values for Canada, Brazil, and Australia (Figures and Tables S11), because these countries are among the top four adopters of CA (see Table S7). Although this provides a comparison to independent data, it
cannot be considered as a validation because of temporal mismatch among compared datasets and aggregation uncertainty when using Global Administrative Areas (2015) for aggregating tillage areas to sub-national scale. For each of the selected countries our downscale algorithm can quite well reproduce the main no-tillage area but tends to allocate too much CA area to certain regions instead of a more homogenous spread, which spatial pattern can be rather deduced from the associated reference maps.

Prestele et al. (2018) state their potential CA area to be 11.3 Mkm² in their 'Bottom–up' and 5.33 Mkm² in their 'Top-down' scenarios until the year 2050. Our estimate of potential CA-suitable area of 4.66 Mkm² is lower but of the same magnitude as of their 'Top-down' scenario, despite the differing assumptions and using a slightly different CA mapping approach. Prestele et al. (2018) used a different cropland product and targeted another time period, than we did also resulting in slight area deviations compared to our derived suitable and potentially
suitable CA area. We decided to produce our tillage dataset and source code flexible in the way that each modeling group may adjust it to their own default cropland input.

### 4.2    Potentials, limitations, and implications for applications of the dataset

Agricultural land management practices are not only determined by environmental factors, but are embedded in local to regional systems of culture, traditions, and markets. This mosaic of farming conditions can only be taken
into account at high spatial resolution. The developed tillage dataset is an attempt to better account for heterogeneous patterns of agricultural management across and within countries by using socio-economic and



biophysical data in conjunction. The resolution of the generated dataset with 0.083° is quite high, while most global ecosystem models currently run on 0.5° resolution and may have to aggregate the data for input usage.

A limitation to our presented mapping approach is that the input datasets applied cover different time periods,
e.g. GLADIS reports water erosion values for approximately the year 2000, SPAM2005 and the field size dataset for the year 2005, the aridity spans to the reference climate data of the period from year 1961 to1990, and for some countries we extracted the only CA reporting year by FAO (2016) from years 2002 up to 2013. By using SPAM2005, field size for 2005, and setting the objected year for the produced tillage dataset to 2005 as well, we tried to minimize inconsistencies in time coverage at least for the cropland data.

The tillage dataset presented here can be assumed to be employed in various applications, depending on the type of model, context, and objective of the user. A challenge of the full usage of this dataset is the limited implementation of the 42 crop types reported in SPAM2005 in global ecosystem models. Especially perennial crop types are hardly ever parameterized in global biophysical models or if so are rather addressing regional scale application (Fader et al., 2015). One reason for the missing implementation may be their relatively small
cultivation areas globally (~10% of global cropland (Erb et al., 2016)). Woody and other perennial plant species entail interesting potential in the aspect of sustainable agricultural practices because they keep the soil covered for longer periods and thus better protect it from erosive and radiative forces, promote soil organic carbon accumulation (Smith et al., 2008), and stabilize soils more than annually planted crop types.

Another challenge for the application of our tillage dataset in model simulations is the differentiation of soil
depth affected by the tillage operation. Some models may be able to differentiate between 20 or 30 cm depth affected by the tillage operation mostly when having a site-based background and therefore a very detailed representation of agricultural management practices (White et al., 2010). The global dynamic ecosystem model LPJ-GUESS and the Community Land Model (CLM) have implemented the tillage routines as a tillage factor accelerating the decomposition rate of the different soil carbon pools (Levis et al., 2014; Olin et al., 2015), so
that implementations of spatial variability in depth or mixing efficiency are not straight forward.

White et al. (2010) elaborate on the problem of generally implementing a three dimensional aspect as "surface affected" by the tillage practice, which would be the case for simulating reduced tillage practices as strip-, mulch-, or ridge-till, weed management during the growing period of the main crop, or for preparing the seedbed for inter-cropping cultures. The reduction relates to depth, surface affected or both, for which White et al. (2010)
recommend an intermediate model implementation mode which distinguishes two zones, as one share of the soil being affected and the other one not.

Some authors mention partial adoption of CA as referring to the minimal soil disturbance practice only (Giller et al., 2015; Scopel et al., 2013) where residues are not always retained (Pittelkow et al., 2015). This no-tillage practice tries to benefit from saving energy, work hours, machine wearing, and field passes when skipping
tillage. No-tillage without a sufficient biological mulch is reliant on the application of increased amounts of herbicides to comply with weeds (McConkey et al., 2012; Mitchell et al., 2012) compared to conventional tillage systems. Leaving the soil unprotected, exposes the soil surface to erosive forces, and enhances nutrient leakage especially under high rainfall intensities. Crusting and compaction of the soil can only be addressed by tilling these fields rotationally, as has been discussed in Erb et al. (2016). This rotational tillage may lead to a decrease
of soil organic matter (SOM) due to increased mineralization under aerated conditions and the advantages of not-tilling during the other years disappears (Powlson et al., 2014). The effects of SOM increase under no-tillage



only in conjunction with a certain amount of residue inputs, may appear relevant after a transition time of about 10 to 20 years of continuous practice until a new equilibrium state of SOM dynamics is re-established (Sá et al., 2012). The other often missing aspect to the full implementation of the CA practice is the rotation of diverse

crop types, inter-cropping or other green manuring practices. It remains unclear to what extent countries reporting CA area to FAO may rather refer to this partial adopted practice of CA.

The tillage dataset is a major step forward compared to globally rather homogeneous assumptions on tillage systems (Hirsch et al., 2017; Levis et al., 2014) or a total ignorance of soil management practices (Folberth et al., 2016; Rosenzweig et al., 2014). The rule-based approach and the publication of the underlying data processing

scripts allow for extensions of this work, if further relationships can be identified or improved data become available. It also allows for constructing future scenarios, consistent with other scenario frameworks on climate, economic development, and land-use change (e.g. Popp et al. (2017)). Further research is needed to generate land management datasets with high resolution on crop rotations, residue management, and multiple cropping, so that the full set of CA principles can be simulated and biophysically assessed in comparison to further

sustainable land practices.

## 5    Data availability

The presented tillage system dataset and source code are accessible via an open-data repository for modeling communities interested in the quantitative assessment of biophysical and biogeochemical impacts of land use and soil management. The tillage dataset can be downloaded from: http://doi.org/10.5880/PIK.2018.012 and the

corresponding R-code from: http://doi.org/10.5880/PIK.2018.013. Supplementary information is available in the online version of this article.

## Author  contributions

V. Porwollik, C. Müller, S. Rolinski, and J. Heinke developed the tillage system dataset. V.P. collected the input data and wrote the scripts for processing and analyzing the data. C.M and J.H. suggested the CA area

downscaling procedure whereas J.H. proposed the application of logit model. V.P. prepared the manuscript with contributions from all co-authors with respect to interpretation of the results and writing of the final paper.

## Competing interests

The authors declare that they have no conflict of interest.

## Acknowledgements

S.R. and V.P. acknowledge financial support from the MACMIT project (01LN1317A) and J.H. from the SUSTAg project (031B0170A) both funded through the German Federal Ministry of Education and Research (BMBF). We thank Jannes Breier (PIK) for support with data processing in R and Steffen Fritz (IIASA) and Theodor Friedrich (FAO) for personal communication.





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



**Table 1** Six tillage systems and suggested parametrization for model applications (note that a) several values per tillage system refer to each single tillage event within each tillage system in the same order as mentioned under the frequency per year, and b) for reduced tillage the inversion and mixing efficiency is depending on the specific form of reduced tillage as mentioned above).

| Tillage system | Conventional annual tillage | Rotational tillage | Conservation Agriculture | Traditional annual tillage | Traditional rotational tillage | Reduced tillage |
|---|---|---|---|---|---|---|
| **Soil management components** | Tillage for seedbed preparation, cultivation, post-harvest tillage | Tillage for seedbed preparation, cultivation, post-harvest tillage | Minimum mechanical soil disturbance with direct seeding | Hoe or cutlass for seedbed preparation, cultivation, post-harvest tillage | Hoe or cutlass for seedbed preparation, cultivation, post-harvest tillage | Tillage for seedbed preparation, cultivation, post-harvest tillage |
| **Soil layer inversion per tillage operation** | Yes, no, yes | Yes, no, yes | No | Yes, no, yes | Yes, no, yes | (Yes), no, (yes) |
| **Frequency and timing per year** | 1 before seeding, 1 to 2 cultivation (10 days to 2 weeks after establishment), 1 after harvest | 1 before seeding, annually 1 to 2 cultivation, 1 after removal | 1 at seeding | 1 before seeding, 1 to 2 cultivation (10 days to 2 weeks after establishment), 1 after harvest | 1 before seeding, annually 1 to 2 cultivation, 1 after removal | 1 before seeding, 1 to 2 cultivation (10 days to 2 weeks after establishment), 1 after harvest |
| **Depth (cm)** | 20, 5, 20 | 20, 5, 20 | 5 | 10, 5, 10 | 10, 5, 10 | <15, 5, <15 |
| **Mixing efficiency (%)** | 90, 20, 90 | 90, 20, 90 | 5 | 50, 20, 50 | 50, 20, 50 | 90, 20, 90 |
| **Soil surface affected (%)** | 100, 33, 100 | 100, 33, 100 | 20 -25 | 100, 33, 100 | 100, 33, 100 | 100, 33, 100 |
| **Soil surface covered by residues after seedbed preparation (%)** | <15 | <15 | >30 | <15 | <15 | 15 -30 |






**Table 2** Gridded and national scale datasets used for mapping tillage

| Global gridded dataset | Resolution (degree) | Temporal coverage (year) | Source |
|---|---|---|---|
| Crop-specific cropland | 0.083° | 2005 | SPAM2005: IFPRI/IIASA (2017b) |
| Soil depth to bedrock | 0.1° | 1990-2014 | SoilGrids: Hengl et al. (2014) |
| Field size | 0.0083° | 2005 | Fritz et al. (2015) |
| Water erosion | 0.083° | 1990-2011 (~2000) | GLADIS: Nachtergaele et al. (2011) |
| Aridity | 0.16667° | 1961-1990 | FAO (2015) |
| **National data** | | | |
| Conservation Agriculture (CA) area | country | 2002-2013 | FAO (2016) |
| Income level | country | 2005 | World Bank (2017) |

**Table 3** Correlation coefficients (r) according to 'Pearson' between spatial predictor variables (crop mix, field size, erosion, and aridity) across all grid cells containing CA-suitable cropland globally.

| (r) | Field size | Erosion | Aridity |
|---|---|---|---|
| Crop mix | 0.322 | -0.104 | -0.241 |
| Field size | | -0.356 | -0.141 |
| Erosion | | | -0.002 |


**Table 4** Logit model input parameters, as midpoint ($xmid$) and slope ($k$) of the four logit model input datasets (columns 1 and 2), which are altered per sensitivity setting. Correlation coefficients ($r$) for ranks according to 'Spearman' between the reference case (Logit-ref) and the perturbed logit model version results are given, illustrating the sensitivity of the grid cell likelihood to have CA-suitable area (columns 3 to 6).

| Variable | Logit-ref (xmid) | Logit-ref (k) | Logit-ref and k+100 % (r) | Logit-ref and k-50 % (r) | Logit-ref and drop one variable (r) | Logit-ref and one variable only (r) |
|---|---|---|---|---|---|---|
| Field size | 20 | 0.250 | 0.975 | 0.988 | 0.944 | 0.555 |
| Erosion | 12 | 0.017 | 0.992 | 0.997 | 0.989 | -0.119 |
| Aridity | 0.650 | -5 | 0.966 | 0.982 | 0.901 | 0.607 |
| Crop mix | 0.500 | 10 | 0.981 | 0.971 | 0.773 | 0.826 |


**Table 5** Global sums over 22 CA-suitable crop type areas (ha), share of downscaled CA area values on the identified CA-suitable area (%), and crop-specific downscaled CA areas (ha).



| Crop type | Area suitable for CA (ha) | Share of downscaled on area suitable for CA (%) | Downscaled CA area (ha) |
|---|---|---|---|
| Soybean | 74,085,533 | 48 | 35,922,509 |
| Wheat | 134,155,907 | 24 | 32,123,029 |
| Maize | 76,236,593 | 19 | 14,345,219 |
| Barley | 48,540,127 | 12 | 5,798,762 |
| Rape | 14,463,189 | 31 | 4,536,453 |
| Sunflower | 18,626,706 | 20 | 3,672,963 |
| Sorghum | 9,784,525 | 24 | 2,380,535 |
| Bean | 11,986,897 | 20 | 2,355,186 |
| Other cereals | 23,138,040 | 10 | 2,211,589 |
| Cotton | 8,408,017 | 25 | 2,112,172 |
| Other pulses | 7,685,206 | 21 | 1,595,015 |
| Lentils | 1,901,924 | 45 | 856,723 |
| Pearl millet | 5,601,798 | 11 | 588,589 |
| Rest | 8,208,157 | 5 | 407,596 |
| Groundnut | 4,722,927 | 7 | 330,506 |
| Chic pea | 2,847,020 | 11 | 322,613 |
| Small millet | 1,341,620 | 21 | 286,056 |
| Vegetables | 9,053,627 | 2 | 183,537 |
| Tobacco | 1,367,825 | 7 | 91,765 |
| Sesame seed | 1,795,517 | 3 | 49,984 |
| Pigeon pea | 638,036 | 2 | 15,150 |
| Cowpea | 632,054 | 1 | 4,812 |
| **World** | **465,221,244** | **24** | **110,190,763** |

**Table 6** Global aggregated cropland area (ha) and share (%) per tillage system

| Tillage system | Sum over cropland and grid cells (ha) | Share of tillage system area on total SPAM2005 cropland (%) |
|---|---|---|
| Rotational tillage | 74,218,834 | 6.56 |
| Traditional rotational tillage | 65,044,354 | 5.75 |
| Traditional annual tillage | 401,538,934 | 35.49 |
| Conservation Agriculture | 110,190,763 | 9.74 |
| Conventional annual tillage | 465,037,862 | 41.10 |
| Reduced tillage | 15,407,865 | 1.36 |
| **World** | **1,131,438,612** | **100** |



**Table 7** Derived tillage system area results compared to estimates of Erb et al. (2016) on tillage intensity areas. The first two columns show our aggregated tillage system area values, columns three and four additionally include the young and temporal fallow cropland area by Siebert et al. (2010), a cropland area not represented in SPAM2005 and therefore added to our total cropland as well as to the 'low intensity' category as described in Erb et al. (2016). Note that Siebert et al. (2010) state, that about 440,000,000 ha of cropland were young and temporal fallow (< 5 years) around the year 2000.

| Tillage system group | Tillage area this study (ha) | Tillage area this study (%) | Tillage area this study + fallow (ha) | Tillage area this study + fallow (%) | Tillage area (ha) (Erb et al., 2016) | Tillage area (%) (Erb et al., 2016) |
|---|---|---|---|---|---|---|
| **Low intensity** | 264,861,816 | 23.4 | 704,861,816 | 44.9 | 473,000,000 | 38.9 |
| **High intensity** | 866,576,796 | 76.6 | 866,576,796 | 55.1 | 743,000,000 | 61.1 |
| **World** | 1,131,438,612 | 100 | 1,571,438,612 | 100 | 1,216,000,000 | 100 |