# Peer review of "Generating a global gridded tillage dataset"

_Earth System Science Data, 2018_

## Referee Comment (RC7)

**Major comments**

The paper titled "Generating a global gridded tillage dataset" by Vera Porwollik et al. reports on the generation of a global classification of tillage practices and the spatial mapping of crop specific tillage systems for the year 2005. This paper provides a new dataset that could potentially be useful in evaluating soil erosion and carbon balance in climate change studies. Furthermore, this paper provides a first global tillage dataset but also R script to estimate the dataset. This dataset is unique and can be applied to a future climate model in order to evaluate climate change and artificial warming effect in the present climate. Overall, I recommend that it should be published after a few minor revisions. My main concerns are as follows:

(1) This paper does not adhere to a single unit to describe area. The abstract and discussion use "$Mkm^2$" while all figures and tables use "ha". Readers cannot compare the different units directly and need to convert "ha" to "$Mkm^2$". Thus, I recommend that the authors use "$km^2$" instead of "ha" in all the figures and tables. The main document should be also modified accordingly.

(2) I tried to run R script on my PC; however it did not run because it required some data to run in my environment. Thus, I recommend that sample R script should be provided with sample input dataset and output dataset. Then reader can run R script with the data provided and verify their output against a sample output dataset that can be involved. Otherwise, the reader cannot check whether their results were correctly reproduced or not.

(3) In terms of figure 1, I suppose that each box in the figure should correspond to the R script. Thus, it is better to add information (ex. line number) that indicates which part of the R script corresponds to the boxes, and also to show the location in the R script, where different crop-specific tillage systems are evaluated. According to these, readers can easily understand the R script, which is also the authors' objective.

---

## Referee Comment (RC1) · Anonymous Referee #1 · 4 Jan 2019

- the paper is well researched and written and the authors display a good understanding of the subject matter and know the relevant literature. - the subject is of interest, new and relevant - comment on line 266: the Pittelkow data are not reliable source, since they are derived from a metanalysis of not accurate data; practical field experiences particularly in rice, but also in some root and tuber crops (cassava, portato) show same or higher yields under no till and no puddling. - in general the wording "land suitable for CA" should be changed. There is no land which is not suitable for sustainable farming, but those land areas referred to as suitable might be more likely for adoption of CA while other land or crop areas might require more assistance or support for adoption. In general a very good paper.

---

## Referee Comment (RC2) · Anonymous Referee #2 · 8 Jan 2019

General comments The paper is well constructed, thought out, written and presented. Only minor revisions are need.

Specific comments The paper would be improved if the "bigger picture" was considered. In the discussion the authors discuss the impact of the work and the use the dataset could be put to but it would be an improvement to see this in the abstract and introduction. If you want the readers to use the data then you need to promote its uses as early as possible.

Section 5 "Data Availability" needs expanding. Although you provide links to the data repository it would be an improvement to give some details of the structure of the data files. You seem to be using netCDF but it is worthwhile telling the reader the "flavour" of the format: are you using netCDF3, netCDF4 netCDF4 - classic for example. The other point is are these file CF compliant and if so which standard you are working to.

It is always useful to the potential user to know what meta data (global attributes) are in the files and if the file naming structure has any useful information embedded in it. It's also useful to provide the reader with an indication of what variables are in the files etc. The final point is about the user licence and if the data set has a DOI.

Technical corrections Table 1: increase column width to allow "Conservation" to appear on one line Table 5: Increase column width to make "Logit-ref\and k-50%" to appear as "Logit-ref and\k-50%" - make the column title structure consistent between columns Tables general: consider using central justification as it will improve the appearance.

Line 221: Sentence "We developed several rules have been in order.." does not make sense. Suggest removing "have been" Line 225: Replace "to" with "of" Line 262: Change "most efficient and homogeous" to "more efficiently and homogeneously" Line 297: Change "few" to "low" Line 454: Change "It is" to "It has" Line 455: Change to "South of the Sahal region"

Figure 1: The diagram is ok as it stands but would be much improved is standard flow chart practices were followed and the "yes\no" decisions were added to the relevant lines.

---

## Referee Comment (RC3) · Anonymous Referee #3 · 10 Jan 2019

GENERAL COMMENTS:

Interesting paper, well written and structured, easy to follow, innovative dataset and relevant for many applications.

My main concern is that the authors have used a series of assumptions and simplified rules to produce their deterministic dataset. However, they haven't acknowledged the uncertainties derived from this process. How confident the user can be in the categories assigned to each cell?

I understand that a partial or full verification is not feasible due to the lack of verification data. As the authors mention, the figures/table in S11 can't be considered as a verification as there is a mismatch in the dates. However, the results do suggest that there can be large errors locally.

Also, there is no mention of uncertainties in the input dataset used (point 2.2.). How reliable are the input datasets used and how is this going to influence the output dataset?

All this should be more explicitly acknowledged in your discussion, so that users are fully aware of the limitations of the dataset. This is my main criticism which I would like to see addressed.

However, other than that, I found the paper very interesting, pleasant to read and pertinent, and I only have a few additional minor corrections to suggest.

SPECIFIC COMMENTS:

Figure 2: In general, the figures/maps are nice and the choice of colour palette is adequate, except for figure 2, which uses the "rainbow" colour scheme. The 'rainbow' palette is the default one in many mapping software, and has been widely used in the past. However, it not only poses problem for colour-blind readers (approx. 10% of male population), but also gives misleading perceptions of thresholds in data (e.g. Light and Bartlein, 2004; Hawkins et al., 2015). There is growing support within the scientific community to abandon the use of rainbow colour scheme. It is of course ultimately a personal choice from the authors, but I would suggest you redo the map choosing a different colour scheme.

Reference:

Light, A. and Bartlein, P. J.: The end of the rainbow? Color schemes for improved data graphics, Eos Trans. Am. Geophys. Union, 85(40), 385–391, doi:10.1029/2004EO400002, 2004.

Hawkins, E., McNeall, D., Williams, J., Stephenson, D. and Carlson, D. (2015) Graphics: scrap rainbow colour scales. Nature, 519 (7543). p. 291. ISSN 0028-0836 doi: https://doi.org/10.1038/519291d

Paragraph page 18, line 497-502: Any hypothesis or explanation as to why your results are different from the findings of Erb et al. (2016)? You provide short interpretations to

the other comparisons you have made. Please add something here too.

TECHNICAL CORRECTIONS:

Page 6, line 222: remove "have been"

Page 13, line 410: remove "of"

Page 18, line 508: remove "in" in "may persist low in"

---

## Referee Comment (RC4) · Anonymous Referee #4 · 11 Jan 2019

I have read the manuscript with much interest to understand the importance of the work and if it really fills a gap in our knowledge. Reading the manuscript has not been much easy because it is too complex because of both the way of presenting the topic and the proposed methods. Soil tillage is an important research issue for its effects on soil conservation and carbon sequestration but the described approach at global scale is not much suitable to help in quantitative assessment of biophysical and bio-geochemical impacts of land use and soil management as claimed by Authors. They have pointed out clearly the many factors and properties, which can determine the type of soil tillage. Among these are included soil type and depth, climate, crops, rainfed and irrigated crops, socio-economic factors determining the mechanization level of agriculture, etc. Consequently, it results extremely complex and difficult to model all factors and properties. Particularly, the Authors have used data much different that have required to be resampled and aggregated (Line 220) but no detail has been provided on

how that has been made. Many rules have been used for mapping and downscaling but it is not much clear how the Authors have statistically validate them.

The manuscript should be organized better to allow readers to follow the development of the objectives in materials, methods, and results. The quality of writing should be checked and improved. There is an excessive use of first person: we .... The title should be made more effective and to reflect better the objectives. The abstract should summarize better the whole manuscript. The Introduction section should be made more fluent and readable. The novelty should be explained better and the objectives made clearer. Methods should be organized better to allow readers understanding how methods have been used. Too short subsections should be merged. Results and Discussion sections would require to be supported by improved Methods and data section.

---

## Referee Comment (RC6) · Anonymous Referee #5 · 23 Jan 2019

The paper presents a global tillage dataset by generating a clear classification of tillage practices and mapping the crop-specific tillage systems and the probability of cropland areas suitable for conversation agriculture at a resolution of 5 arc-minutes. As an improvement relative to previous studies cited in the paper, this dataset used water erosion, aridity, field size, and crop mix data per grid cell as spatial predictors to determine the distribution of national reported conversation agriculture area within a country in a logit model. The high-resolution map of tillage can improve modeling the soil carbon cycle in global Earth system models. The presentation is almost clear, but the English can be improved. I recommend publication of this paper in Earth Syst. Sci. Data. There are some minor comments below.

Specific comments:

Line 58: What is HYDE?

Line 60: "For downscaling national values Prestele et al. (2018) …" this sentence is too complicated. Should be rephrased.

Line 94: What is ESM?

Line 106: I do not understand the sentence "… or can assess different tillage impacts just in form of scenarios". Should be rephrased.

Line 110: "increase understanding of the drivers for different tillage practices". What do the authors mean by "drivers"?

Line 222: "We developed several mapping rules have been in order to allocate the…" this sentence is too complicated. Should be rephrased.

Line 228: Here the authors mentioned the depth of 15 cm, but claimed that "we decided for a minimum depth of mechanized tillage of 20 cm" above. Please explain this inconsistency (the same for Figure 2).

Line 533: "global ecosystem models currently run on 0.5° resolution and may have to aggregate the data for input usage" this is not always the case. In many ecosystem models (e.g. ORCHIDEE), their dynamics are simulated at a coarse resolution, but they divide the large model pixel to smaller ones in considering the agricultural processes.

---

## Author Comment (AC1) · 18 Mar 2019

Authors' responses to comments - referee #1

Thank you very much for providing helpful feedback. Please find below a point by point response to your comments.

Referee #1: Comment on line 266: the Pittelkow data are not reliable source, since they are derived from a metanalysis of not accurate data; practical field experiences particularly in rice, but also in some root and tuber crops (cassava, portato) show same or higher yields under no till and no puddling.

Authors' response: Thank you for pointing out this uncertainty in assumptions made building on data of Pittelkow et al. (2015). We improved our statement in the text, first by shifting the paragraph to the section 2.4.1 describing the concrete CA area

downscaling to avoid confusion on the mapping rules described in the other tillage system area derivation.

Further we add on the feasibility of applying no-tillage for rice and roots/tubers production as follows: All annual rainfed root, tuber and rice cropland is excluded from the potential CA area following Pittelkow et al. (2015), who reported larger yield penalties for these crop types when applying no-tillage practices. Rice is often produced as paddy rice, requiring puddling, which is a practice modifying the soil aggregates a lot in order to facilitate the flooded condition, e.g. to suppress weed growth. A conversion from puddled to dryland rice production as well as improved drainage of tuber crops production area may require additional management steps by the farmer in order to achieve comparable yield levels with no-tillage as under conventional production methods.

Referee #1: In general the wording "land suitable for CA" should be changed. There is no land which is not suitable for sustainable farming, but those land areas referred to as suitable might be more likely for adoption of CA while other land or crop areas might require more assistance or support for adoption.

Authors' response: We support your argument that theoretically all croplands can be farmed in a sustainable way. In the manuscript and R-script we change 'land suitable for CA' and 'potentially CA-suitable area' to 'potential CA area' as wordings also used in Prestele et al. (2018) and 'scenario CA area' respectively.

References

Pittelkow, C. M., Linquist, B. A., Lundy, M. E., Liang, X., van Groenigen, K. J., Lee, J., van Gestel, N., Six, J., Venterea, R. T., and van Kessel, C.: When does no-till yield more? A global meta-analysis, Field Crops Research, 183, 156-168, doi: 10.1016/j.fcr.2015.07.020, 2015.

Prestele, R., Hirsch, A. L., Davin, E. L., Seneviratne, S. I., and Verburg, P. H.: A spatially explicit representation of conservation agriculture for application in global change studies, Global Change Biology, 24, doi:10.1111/gcb.14307, 2018.

---

## Author Comment (AC2) · 18 Mar 2019

Thank you very much for providing helpful feedback. Please find below a point by point response to your comments.

Referee #2: The paper would be improved if the "bigger picture" was considered. In the discussion the authors discuss the impact of the work and the use the dataset could be put to but it would be an improvement to see this in the abstract and introduction. If you want the readers to use the data then you need to promote its uses as early as possible.

Authors' response: Thank you for this suggestion. We will modify the abstract and introduction section, already there emphasizing the possible applications and significance of the tillage data set for impact assessment of agricultural practices on carbon, water, and nutrient cycling.

Referee #2: Section 5 "Data Availability" needs expanding. Although you provide links to the data repository it would be an improvement to give some details of the structure of the data files. You seem to be using netCDF but it is worthwhile telling the reader the "flavour" of the format: are you using netCDF3, netCDF4 netCDF4 - classic for example. The other point is are these file CF compliant and if so which standard you are working to It is always useful to the potential user to know what meta data (global attributes) are in the files and if the file naming structure has any useful information embedded in it. It's also useful to provide the reader with an indication of what variables are in the filesetc. The final point is about the user licence and if the data set has a DOI.

Authors' response: We have now extended the data availability section with additional technical details. The user may also refer to more details described on the website and accompanying meta-data of the repository where the code and data set are available for download.

We exchange the text in section 5 with the following: The presented tillage system area dataset and source code are available under the ODBL (data) and MIT (source code) licenses. The tillage area dataset can be downloaded from: http://doi.org/10.5880/PIK.2018.012 and the corresponding R-code from: http://doi.org/10.5880/PIK.2018.013. The dataset is provided in netCDF format (version 4) and consists of 42 layers each reporting crop-specific tillage types per grid cell. Additionally, we provide a layer with indicating area to where CA may be likely (scenario CA area). The dataset can also be applied as a mask or overlay for identifying tillage area. The R-code is provided to enable other modelling groups to adjust our tillage area mapping algorithm to their needs, e.g. for different input data or scenarios. Supplementary information (SI) is available in the online version of this article.

Referee #2: Technical corrections Table 1: increase column width to allow "Conservation" to appear on one line

[Figure]

Authors' response: Agree.

Referee #2: Table 5: Increase column width to make "Logit-ref\and k-50%" to appear as "Logit-ref and\k-50%" - make the column title structure consistent between columns

Authors' response: Agree.

Referee #2: Tables general: consider using central justification as it will improve the appearance.

Authors' response: Agree.

Referee #2: Line 221: Sentence "We developed several rules have been in order.." does not make sense.

Authors' response: Agree and we changed the sentence to: We developed several rules in order to allocate the derived tillage systems to the grid scale.

Referee #2: Line 225: Replace "to" with "of"

Authors' response: Maybe there is a misunderstanding but we improved the sentence by deleting the ending "s" in "units" resulting in the following formulation: "...to distribute data of a larger spatial unit to the grid cell level..."

Referee #2: Line 262: Change "most efficient and homogeous" to "more efficiently and homogeneously"

Authors' response: Agree and improved to: ... because efficient and equal distribution of water requires some leveling off of the field to flatten the surface in order to distribute irrigation water more efficiently and homogeneously over the field.

Referee #2: Line 297: Change "few" to "low"

Authors' response: Agree.

Referee #2: Line 454: Change "It is" to "It has"

[Figure]

Authors' response: Agree.

Referee #2: Line 455: Change to "South of the Sahal region"

Authors' response: We exchanged the former formulation to the following sentence: It occurs in Mexico, South of the Sahel region but mostly is found on cropland in India (Table S8 for further metrics across tillage system areas; Table S9).

Referee #2: Figure 1: The diagram is ok as it stands but would be much improved is standard flowchart practices were followed and the "yes\no" decisions were added to the relevant lines.

Authors' response: We adjusted slightly the settings of the flow chart but refrained from adjusting for exact flow chart standard as diamond shape for process (decisions) would require more space than our chosen rectangle shape but we add the "yes\no" decisions to the relevant lines and updated figure 1 in the manuscript.
* * *

---

## Author Comment (AC3) · 18 Mar 2019

Thank you very much for providing helpful feedback. Please find below a point by point response to your comments.

Referee #3: My main concern is that the authors have used a series of assumptions and simplified rules to produce their deterministic dataset. However, they haven't acknowledged the uncertainties derived from this process. How confident the user can be in the categories assigned to each cell? I understand that a partial or full verification is not feasible due to the lack of verification data. As the authors mention, the figures/table in S11 can't be considered as a verification as there is a mismatch in the dates. However, the results do suggest that there can be large errors locally.

Authors' response: Our mapping rules are generated on the basis of literature findings on globally relevant tillage types, their underlying reasons, and purposes. In the

absence of any statistical data for soil management at the global scale (except for Conservation Agriculture (CA) practices), we use proxy relations and data which can indicate tillage types of relevant difference but representative enough for existing cropping systems. We are aware that the use of proxy data and an area prioritization based on simple rules cannot reproduce the spatial patterns of actual tillage systems but rather should be seen as an approximation to reality making best use of available knowledge and data. The comparisons to other data illustrate that mayor spatial patterns can be reproduced but as you mentioned locally errors might be large. We have extended the discussion of these points in section 4.2, making clear that the data set presents a scenario of current conditions that is based on plausible combinations of best knowledge and data.

Referee #3: Also, there is no mention of uncertainties in the input dataset used (point 2.2.). How reliable are the input datasets used and how is this going to influence the output dataset?

Authors' response: Indeed, each input data set comes with its own uncertainties, which is often not described explicitly but reflected in discrepancies between different data sets on the same entity such as land use patterns (see e.g. Porwollik et al. 2017). We have not explicitly tested the propagation of input data uncertainty but focused on the uncertainty in the parametrization of our allocation rules. We now include this aspect in the discussion, as suggested.

Referee #3: All this should be more explicitly acknowledged in your discussion, so that users are fully aware of the limitations of the dataset. This is my main criticism which I would like to see addressed.

Authors' response: We will add text on uncertainty of our rule based approach, the used input and the output data.

Referee #3: Figure 2: In general, the figures/maps are nice and the choice of colour palette is adequate, except for figure 2, which uses the "rainbow" colour scheme. The

'rainbow' palette is the default one in many mapping software, and has been widely used in the past. However, it not only poses problem for colour-blind readers (approx. 10% of male population), but also gives misleading perceptions of thresholds in data (e.g. Light and Bartlein, 2004; Hawkins et al., 2015). There is growing support within the scientific community to abandon the use of rainbow colour scheme. It is of course ultimately a personal choice from the authors, but I would suggest you redo the map choosing a different colour scheme.

Authors' response: We agree to your suggestion and improved figure 2 by dropping the rainbow but applying the viridis-color scheme with a break per color step. We additionally included a further break point resulting in an increased shaded pattern in the probability map (in what has been shaded all red only, now is appearing in yellow to light greenish colors). These finer scaling shows more clearly that a lot of high probability values end up in between 0.9 and 1 but especially a lot between 0.999 and 1.

Referee #3: Page 6, line 222: remove "have been"

Authors' response: Agree.

Referee #3: Page 13, line 410: remove "of"

Authors' response: Agree.

Referee #3: Page 18, line 508: remove "in" in "may persist low in"

Authors' response: Agree.
* * *

---

## Author Comment (AC4) · 18 Mar 2019

Thank you very much for providing helpful feedback. Please find below a point by point response to your comments.

Referee #4: I have read the manuscript with much interest to understand the importance of the work and if it really fills a gap in our knowledge. Reading the manuscript has not been much easy because it is too complex because of both the way of presenting the topic and the proposed methods. Soil tillage is an important research issue for its effects on soil conservation and carbon sequestration but the described approach at global scale is not much suitable to help in quantitative assessment of biophysical and bio-geochemical impacts of land use and soil management as claimed by Authors. They have pointed out clearly the many factors and properties, which can determine the type of soil tillage. Among these are included soil type and depth, climate, crops, rain-

[Figure]

fed and irrigated crops, socio-economic factors determining the mechanization level of agriculture, etc. Consequently, it results extremely complex and difficult to model all factors and properties.

Author's response: Indeed, the decisions made by farmers on which type of tillage to use are complex, substantially more complex than reflected by our rules. We will revise the structure and text of the article where suitable to improve the readability of the article. We do think that providing an explicit data set on tillage types is helpful in the quantitative assessment of biophysical and biogeochemical effects of land use and soil management, as the alternative is to use implicit model assumptions.

Referee #4: Particularly, the Authors have used data much different that have required to be resampled and aggregated (Line 220) but no detail has been provided on how that has been made.

Author's response: Until now we refrained from describing too many technical details concerning the coding as we thought that would blow up and complicate the text even more. For us it was more important to explain the general concept. Indeed, substantial harmonization steps of data formats were necessary to process the different data sources. We have now expanded the description of theis harmonization procedure. Full detail on the data data processing steps is also provided through the accompanying published source code (Porwollik et al., 2018).

Referee #4: Many rules have been used for mapping and downscaling but it is not much clear how the Authors have statistically validate them.

Author's response: We derived rules from qualitative statements found in relevant literature (for CA – erosion, CA- aridity, and CA- crop type, the threshold of 2 ha per ha to distinguish between small and large scale farming). For downscaling CA rather to large than to small field size we approved of the relation between CA area and farm size found via a statistical assessment shown in Figure S3 with the coefficient of determination $r^2$=0.66. Further prove of statistical relations among mapping variables

definitely are an interesting challenge to be explored but are momentary outside the scope of this mapping exercise. In order to capture the uncertainty of the logit model we included the sensitivity test with different variable combinations and functional parameters. In the manuscript we will more explicitly describe which rules are based on qualitative relations found in the literature and the ones we proved statistically.

Referee #4: The manuscript should be organized better to allow readers to follow the development of the objectives in materials, methods, and results. The quality of writing should be checked and improved.

Authors' response: We will revise the manuscript for better streamlining the narrative.

Referee #4: There is an excessive use of first person: we ....

Authors' response: There are different perspectives on the use of active and passive voice in articles. We find that active voice makes articles substantially easier to read, but will reduce the occurrence of 'we-formulations' in the manuscript.

Referee #4: The title should be made more effective and to reflect better the objectives.

Authors' response: We improved the title of the manuscript as following: Generating a rule-based global gridded tillage dataset.

Referee #4: The abstract should summarize better the whole manuscript. Referee #4: The Introduction section should be made more fluent and readable. Referee #4: The novelty should be explained better and the objectives made clearer. Referee #4: Methods should be organized better to allow readers understanding how methods have been used. Referee #4: Too short subsections should be merged. Referee #4: Results and Discussion sections would require to be supported by improved Methods and data section.

Authors' response: We improved the manuscript in terms of structural adjustments and better separation into sections. Further we improve the abstract, introduction, data and method, and the discussion in the course of this review process as suggested by all

referees.

References

Porwollik, V., Rolinski, S., and Müller, C.: A global gridded dataset on tillage area - R-code. V. 1.0., GFZ Data Services. http://doi.org/10.5880/PIK.2018.013, 2018.

---

## Author Comment (AC5) · 18 Mar 2019

Thank you very much for providing helpful feedback. Please find below a point by point response to your comments.

Referee #5: The presentation is almost clear, but the English can be improved.

Author's response: We will carefully check the language and improve the wording and formulations.

Referee #5: Line 58: What is HYDE?

Authors' response: HYDE stands for 'History Database of the Global Environment'. HYDE is an internally consistent combination of historical population estimates and allocation algorithms with time-dependent weighting maps for land use including grassland but also cropland including its irrigated and rainfed shares. We now explain that

abbreviation and have corrected the reference.

We improved the formulation as follows: ...Prestele et al. (2018) mapped reported national values of CA area from Kassam et al. (2015) to cropland from the HYDE database (History Database of the Global Environment; Klein Goldewijk et al. (2017)) for the year 2012.

Referee #5: Line 60: "For downscaling national values Prestele et al. (2018)..." this sentence is too complicated. Should be rephrased.

Authors' response: We agree and improved the paragraph as follows: Based on literature findings, Prestele et al. (2018) developed a CA adoption index per grid cell composed by a set of spatial predictors as aridity, field size, soil erosion, market access, and poverty for downscaling reported national CA area values. Their global map of CA at a spatial grid resolution of 5 arc-minutes is freely available for application in impact assessments in global model simulations.

Referee #5: Line 94: What is ESM?

Authors' response: Thank you for that hint – we have simply overseen to define this abbreviation. At first occurrence of the word 'Earth system model' in our manuscript we now introduced the abbreviation 'ESM'.

Referee #5: Line 106: I do not understand the sentence "... or can assess different tillage impacts just in form of scenarios". Should be rephrased.

Authors' response: We agree that our wording is not very precise so we rephrased the section as following: In the absence of detailed area and tillage type information, the global ecosystem modeling community currently can assess difference of contrasting tillage type impacts just in form of stylized scenarios simulating the effect on the entire cropland area (Del Grosso et al., 2009; Olin et al., 2015; Pugh et al., 2015). One recent exception is the assessment by Hirsch et al. (2018) who assess the effects of an altered albedo from residues used for soil cover on CA areas, using the data of
Prestele et al. (2018).

Referee #5: Line 110: "increase understanding of the drivers for different tillage practices". What do the authors mean by "drivers"?

Authors' response: We have revised that sentence to: The objective of this study is to a) increase understanding of differences in tillage practices at the global scale b) formulate rules to spatially map tillage systems to the grid scale, and c) develop an open source and open data crop-specific tillage system dataset for the parameterization of tillage events and area in global ecosystem models and assessments.

Referee #5: Line 222: "We developed several mapping rules have been in order to allocate the..."this sentence is too complicated. Should be rephrased.

Authors' response: We have revised that sentence to: We developed several mapping rules to allocate the six tillage system to the grid scale, employing a decision tree as shown in Fig. 1.

Referee #5: Line 228: Here the authors mentioned the depth of 15 cm, but claimed that "we decided for a minimum depth of mechanized tillage of 20 cm" above. Please explain this inconsistency (the same for Figure 2).

Authors' response: In section 2.1 we state the findings of Pimental and Sparks (2000) who state the minimum soil depth for agricultural production to be 15 cm, which we used as a first indication of suitability for some kind of tillage. We assume generally less soil depth to bedrock necessary for traditional tillage with hand tools and even less for no-tillage under CA. Following the statement of Kouwenhoven et al. (2002), we regard a minimum depth for conventional annual tillage of 20 cm as necessary for managing perennial weeds. So our reduced tillage system area in fact comprises a relic of mismatches between cropland reported by SPAM2005 but at the same time a shallow soil depth reported by our soil depth to bedrock dataset, where we currently cannot deduce which of the data products is right in the affected grid cells or in the
case both are right – how the affected farmers in these specific cases manage their soil.

We improved figure 1 and the entire calculation for the fraction of rotational tillage crops on soil deeper than 15 but shallower than 20 cm depth to bedrock because of this detected inconsistency. That cropland fraction is now newly allocated to the reduced tillage system. An updated version of the tillage data set and R-script will be provided in the context of this revision process.

Improved text at the end of section 2.1: We applied a downscale algorithm of national reported CA area values on potential CA area (see Fig. 1 box "Downscaling"; see following section for more details). The remaining cropland not being assigned to CA is checked again for soil depth to bedrock. In case it was lower than 20 cm, the cropland was assigned to reduced tillage assuming less depth, frequency, mixing efficiency or alternative cultivation practices. In case of soil depth to bedrock of 20 cm or more the remaining cropland was depending on crop type either mapped to the conventional annual or rotational tillage system following the finding of Kouwenhoven et al. (2002) mentioned above for perennial weed management.

Referee #5: Line 533: "global ecosystem models currently run on 0.5° resolution and may have to aggregate the data for input usage" this is not always the case. In many ecosystem models (e.g. ORCHIDEE), their dynamics are simulated at a coarse resolution, but they divide the large model pixel to smaller ones in considering the agricultural processes.

Authors' response: Thank you for the hint. We see that the sentence was generalizing current spatial resolution in global model simulations too much so we rephrased it.

Improved: Global ecosystem models are currently mostly run at a coarser resolution than our data set's resolution and the tillage data may have to be aggregated in such cases. This could introduce further uncertainty to the area under a certain tillage system. Other models (e.g. ORCHIDEE) are able to account for increased resolution of

agricultural input data by dividing the large model pixel to smaller ones in considering the agricultural processes.

References

Del Grosso, S. J., Ojima, D. S., Parton, W. J., Stehfest, E., Heistemann, M., DeAngelo, B., and Rose, S.: Global scale DAYCENT model analysis of greenhouse gas emissions and mitigation strategies for cropped soils, Global and Planetary Change, 67, 44-50, doi: 10.1016/j.gloplacha.2008.12.006, 2009. Hirsch, A. L., Prestele, R., Davin, E. L., Seneviratne, S. I., Thiery, W., and Verburg, P. H.: Modelled biophysical impacts of conservation agriculture on local climates, Global Change Biology, 24, 4758-4774, doi:10.1111/gcb.14362, 2018. Kassam, A., Friedrich, T., Derpsch, R., and Kienzle, J.: Overview of the Worldwide Spread of Conservation Agriculture. Field Actions Science Reports [Online].5., 2015. Klein Goldewijk, K., Beusen, A., Doelman, J., and Stehfest, E.: Anthropogenic land use estimates for the Holocene – HYDE 3.2, Earth Syst. Sci. Data, 9, 927-953, doi: 10.5194/essd-9-927-2017, 2017. Kouwenhoven, J. K., Perdok, U. D., Boer, J., and Oomen, G. J. M.: Soil management by shallow mouldboard ploughing in The Netherlands, Soil and Tillage Research, 65, 125-139, doi: 10.1016/S0167-1987(01)00271-9, 2002. Olin, S., Lindeskog, M., Pugh, T. A. M., Schurgers, G., Wårlind, D., Mishurov, M., Zaehle, S., Stocker, B. D., Smith, B., and Arneth, A.: Soil carbon management in large-scale Earth system modelling: implications for crop yields and nitrogen leaching, Earth Syst. Dynam., 6, 745-768, doi: 10.5194/esd-6-745-2015, 2015. Pimental, D. and Sparks, D. L.: Soil as an endangered ecosystem, BioScience, 50, 947-947, doi: 10.1641/0006-3568(2000)050[0947:saaee]2.0.co;2 2000. Prestele, R., Hirsch, A. L., Davin, E. L., Seneviratne, S. I., and Verburg, P. H.: A spatially explicit representation of conservation agriculture for application in global change studies, Global Change Biology, 24, doi:10.1111/gcb.14307, 2018. Pugh, T. A. M., Arneth, A., Olin, S., Ahlström, A., Bayer, A. D., Klein Goldewijk, K., Lindeskog, M., and Schurgers, G.: Simulated carbon emissions from land-use change are substantially enhanced by accounting for agricultural management, Environmental Research Letters, 10, 124008,

2015.

---

## Author Comment (AC6) · 18 Mar 2019

Thank you very much for providing helpful feedback. Please find below a point by point response to your comments.

Referee #6: This paper does not adhere to a single unit to describe area. The abstract and discussion use "MkmË̦2" while all figures and tables use "ha". Readers cannot compare the different units directly and need to convert "ha" to "MkmË̦2". Thus, I recommend that the authors use "kmË̦2" instead of "ha" in all the figures and tables. The main document should be also modified accordingly.

Authors' response: We agree and will harmonize all area unit indication in the manuscript to km$^2$.

Referee #6: I tried to run R script on my PC; however it did not run because it required

some data to run in my environment. Thus, I recommend that sample R script should be provided with sample input dataset and output dataset. Then reader can run R script with the data provided and verify their output against a sample output dataset that can be involved. Otherwise, the reader cannot check whether their results were correctly reproduced or not.

Authors' response: It is correct that in order to run the script the user needs to download the input data sets as indicated in our article, R-code but also described in the meta-data at the repository's websites. We will include sample input and output datasets which can be applied for testing the functionality of the R-script.

Referee #6: In terms of figure 1, I suppose that each box in the figure should correspond to the R script. Thus, it is better to add information (ex. line number) that indicates which part of the R script corresponds to the boxes, and also to show the location in the Rscript, where different crop-specific tillage systems are evaluated. According to these, readers can easily understand the R script, which is also the authors' objective.

Authors' response: Thank you for this suggestion. We have now harmonized the wording of the tillage systems between the manuscript, the data-flow diagram (Figure 1) and the accompanying R-script. We improved the structure of the R-script to be more in line with the steps described in the manuscript. In the R-script we also added comments, indicating which section generates which table and values of the manuscript.

---

## Author Response (AR1)

We thank all referees for the careful and positive reviews and comments. Kindly find in the following the point by point responses by the authors to referees' comments and notes and further improvements of the manuscript, accompanying tillage dataset and corresponding R-code (now version 1.1).

| Referee comments: | Author's responses | Improved text |
|---|---|---|
| **Referee #1** | | |
| **Comment on line 266: the Pittelkow data are not reliable source, since they are derived from a metanalysis of not accurate data; practical field experiences particularly in rice, but also in some root and tuber crops (cassava, portato) show same or higher yields under no till and no puddling.** | Thank you for pointing out this uncertainty in assumptions made building on data of Pittelkow et al. (2015). We improved our statement in the text, first by shifting the paragraph to the section 2.4.1 describing the concrete CA area downscaling to avoid confusion on the mapping rules described in the other tillage system area derivation. | All annual rainfed root, tuber, and rice cropland is excluded from the potential CA area following Pittelkow et al. (2015), who reported larger yield penalties for these crop types when applying no-tillage practices. Rice is often produced as paddy rice, requiring puddling, which is a practice modifying the soil aggregates a lot in order to facilitate the flooded condition, e.g. to suppress weed growth. A conversion from puddled to dryland rice production as well as improved drainage of tuber crops production area may require additional management steps by the farmer in order to achieve comparable yield levels with no-tillage as under conventional production methods. |
| **In general the wording "land suitable for CA" should be changed. There is no land which is not suitable for sustainable farming, but those land areas referred to as suitable might be more likely for adoption of CA while other land or crop areas might require more assistance or support for adoption.** | We support your argument that theoretically all croplands can be farmed in a sustainable way. In the manuscript and R-script we now changed 'land suitable for CA' and 'potentially CA-suitable area' to 'potential CA area' as wordings also used in Prestele et al. (2018) and 'scenario CA area' respectively. | Entire manuscript |
| **Referee #2** | | |
| **The paper would be improved if the "bigger picture" was considered. In the discussion the authors discuss the impact of the work and the use the dataset could be put to but it would be an improvement to see this in the abstract and introduction. If you want the readers to use the data then you need to promote its uses as early as possible.** | Thank you for this suggestion. We revised the abstract by already there emphasizing the possible applications and significance of the tillage data set for impact assessment of soil management practices on carbon, water, and nutrient cycling. | Abstract |

| | | |
|---|---|---|
| **Section 5 "Data Availability" needs expanding. Although you provide links to the data repository it would be an improvement to give some details of the structure of the data files. You seem to be using netCDF but it is worthwhile telling the reader the "flavour" of the format: are you using netCDF3, netCDF4 netCDF4 - classic for example. The other point is are these file CF compliant and if so which standard you are working to It is always useful to the potential user to know what meta data (global attributes) are in the files and if the file naming structure has any useful information embedded in it. It's also useful to provide the reader with an indication of what variables are in the filesetc. The final point is about the user license and if the data set has a DOI.** | We have now extended the data availability section with additional technical details. The user may also refer to more details described on the website and accompanying meta-data of the repository where the code and data set are available for download | The presented tillage system dataset and source code are available under the ODBL (data) and MIT (source code) licenses. The tillage dataset can be downloaded from: http://doi.org/10.5880/PIK.2019.009 and the corresponding R-code from: http://doi.org/10.5880/PIK.2019.010. The dataset is provided in netCDF format (version 4) and consists of 42 layers each reporting crop-specific tillage systems per grid cell. Additionally, we provide a layer indicating area, where adoption of Conservation Agriculture could be facilitated (scenario CA area). The dataset can be used as a direct input, be applied as a mask or overlay for identifying tillage area. The R-code is provided to increase transparency of our methods but also to enable other modelling groups to adjust our tillage area mapping algorithm to their needs, e.g. for different input data or scenarios. Supplementary information (SI) is available in the online version of this article. |
| **Technical corrections Table 1: increase column width to allow "Conservation" to appear on one line** | We agree and did so. | |
| **Table 5: Increase column width to make "Logit-ref\and k-50%" to appear as "Logit-ref and\k-50%" - make the column title structure consistent between columns** | We agree and did so. | |
| **Tables general: consider using central justification as it will improve the appearance.** | We agree and did so. | |
| **Line 221: Sentence "We developed several rules have been in order.." does not make sense.** | We agree and revised the sentence. | We developed several rules in order to allocate the derived tillage systems to the grid scale. |
| **Line 225: Replace "to" with "of"** | Maybe there is a misunderstanding but we improved the sentence by deleting the ending "s" in "units". | …to distribute data of a larger spatial unit to the grid cell level… |

| | | |
|---|---|---|
| **Line 262: Change "most efficient and homogeous" to "more efficiently and homogeneously"** | We agree and revised the sentence. | …because efficient and equal distribution of water requires some leveling off of the field to flatten the surface in order to distribute irrigation water more efficiently and homogeneously over the field. |
| **Line 297:  Change "few" to "low"** | We agree and did so. | |
| **Line 454:  Change "It is" to "It has"** | We agree and did so. | |
| **Line 455:  Change to "South of the Sahal region"** | We agree and revised the sentence. | It occurs in Mexico, South of the Sahel region but mostly is found on cropland in India (Table S8 for further metrics across tillage system areas; Table S9). |
| **Figure 1: The diagram is ok as it stands but would be much improved is standard flowchart practices were followed and the "yes\no" decisions were added to the relevant lines.** | We adjusted slightly the settings of the flow chart but refrained from adjusting for exact flow chart standard as diamond shape for processes (decisions) would require more space than our chosen rectangle shape but we add the "yes\no" decisions to the relevant lines and updated figure 1 in the manuscript. | Figure 1 |
| **Referee #3** | | |
| **My main concern is that the authors have used a series of assumptions and simplified rules to produce their deterministic dataset. However, they haven't acknowledged the uncertainties derived from this process. How confident the user can be in the categories assigned to each cell? I understand that a partial or full verification is not feasible due to the lack of verification data. As the authors mention, the figures/table in S11 can't be considered as a verification as there is a mismatch in the dates. However, the results do suggest that there can be large errors locally.** | Our mapping rules are generated on the basis of literature findings on globally relevant tillage types, their underlying reasons, and purposes. In the absence of any statistical data for soil management at the global scale (except for Conservation Agriculture (CA) practices), we use proxy relations and data which can indicate tillage types of relevant difference but representative enough for existing cropping systems. We are aware that the use of proxy data and an area prioritization based on simple rules cannot reproduce the spatial patterns of actual tillage systems but rather should be seen as an approximation to reality making best use of available knowledge and data. The comparisons to other data illustrate that mayor spatial patterns can be reproduced but as you mentioned locally errors might be large. We have extended the | Section 4.2 |

| | | |
|---|---|---|
| | discussion of these points in section 4.2, making clear that the data set presents a scenario of current conditions that is based on plausible combinations of best knowledge and data. | |
| **Also, there is no mention of uncertainties in the input dataset used (point 2.2.). How reliable are the input datasets used and how is this going to influence the output dataset?** | Indeed, each input data set comes with its own uncertainties, which is often not described explicitly but reflected in discrepancies between different data sets on the same entity such as land use patterns (see e.g. Porwollik et al. 2017). We have not explicitly tested the propagation of input data uncertainty but focused on the uncertainty in the parametrization of our allocation rules. We now include this aspect in the section 2.3 and 4.2 discussing input dataset uncertainties, as suggested. | Sections 2.3 and 4.2 |
| **All this should be more explicitly acknowledged in your discussion, so that users are fully aware of the limitations of the dataset. This is my main criticism which I would like to see addressed.** | We added text on uncertainty of our rule based approach, the used input and the output data. | Sections 2.2 and 4.2 |
| **Figure 2: In general, the figures/maps are nice and the choice of colour palette is adequate, except for figure 2, which uses the "rainbow" colour scheme. The 'rainbow' palette is the default one in many mapping software, and has been widely used in the past. However, it not only poses problem for colour-blind readers (approx. 10% of male population), but also gives misleading perceptions of thresholds in data (e.g. Light and Bartlein, 2004; Hawkins et al., 2015). There is growing support within the scientific community to abandon the use of rainbow colour scheme. It is of course ultimately a** | We agree to your suggestion and improved Fig. 2 in the manuscript by dropping the rainbow but applying the viridis-color scheme with a break per color step. We additionally included a further break point resulting in an increased shaded pattern in the probability map (in what has been shaded all red only, now is appearing in yellow to light greenish colors). These finer scaling shows more clearly that a lot of high probability values end up in between 0.9 and 1 but especially a lot between 0.999 and 1. | Figure 2 |

| | | |
|---|---|---|
| **personal choice from the authors, but I would suggest you redo the map choosing a different colour scheme.** | | |
| **Page 6, line 222: remove "have been"** | We agree and did so. | |
| **Page 13, line 410: remove "of"** | We agree and did so. | |
| **Page 18, line 508: remove "in" in "may persist low in"** | We agree and did so. | |
| **Referee #4** | | |
| **I have read the manuscript with much interest to understand the importance of the work and if it really fills a gap in our knowledge. Reading the manuscript has not been much easy because it is too complex because of both the way of presenting the topic and the proposed methods. Soil tillage is an important research issue for its effects on soil conservation and carbon sequestration but the described approach at global scale is not much suitable to help in quantitative assessment of biophysical and bio-geochemical impacts of land use and soil management as claimed by Authors.**
**They have pointed out clearly the many factors and properties, which can determine the type of soil tillage. Among these are included soil type and depth, climate, crops, rainfed and irrigated crops, socio-economic factors determining the mechanization level of agriculture, etc. Consequently, it results extremely complex and difficult to model all factors and properties.** | Indeed, the decisions made by farmers on which type of tillage to use are complex, substantially more complex than reflected by our rules. We will revise the structure and text of the article where suitable to improve the readability of the article. We do think that providing an explicit data set on tillage types is helpful in the quantitative assessment of biophysical and biogeochemical effects of land use and soil management, as the alternative is to use implicit model assumptions. | Entire manuscript |
| **Particularly, the Authors have used data much** | Until now we refrained from describing too many | Section 2.3 |

| | | |
|---|---|---|
| **different that have required to be resampled and aggregated (Line 220) but no detail has been provided on how that has been made.** | technical details concerning the coding as we thought that would blow up and complicate the text even more. For us it was more important to explain the general concept. Indeed, substantial harmonization steps of data formats were necessary to process the different data sources. We have now expanded the description of the harmonization procedure in section 2.3. Full detail on the data processing steps is also provided through the accompanying published source code (Porwollik et al., 2019). | |
| **Many rules have been used for mapping and downscaling but it is not much clear how the Authors have statistically validate them.** | We derived rules from qualitative statements found in relevant literature (for CA – erosion, CA- aridity, and CA- crop type, the threshold of 2 ha per ha to distinguish between small and large scale farming). For downscaling CA rather to large than to small field size we approved of the relation between CA area and farm size found via a statistical assessment shown in Figure S3 with the coefficient of determination $r^2$=0.66. Further prove of statistical relations among mapping variables definitely are an interesting challenge to be explored but are momentary outside the scope of this mapping exercise. In order to capture the uncertainty of the logit model we included the sensitivity test with different variable combinations and functional parameters. In the manuscript, section 4.2 we add text discussing more explicitly which rules are based on qualitative or statistical relations found in the literature. | Section 4.2 |
| **The manuscript should be organized better to allow readers to follow the development of the objectives in materials, methods, and results. The** | We revised the entire manuscript for better streamlining the narrative. | Entire manuscript |

| | | |
|---|---|---|
| **quality of writing should be checked and improved.** | | |
| **There is an excessive use of first person: we ....** | There are different perspectives on the use of active and passive voice in articles. We find that active voice makes articles substantially easier to read,. We reduced the occurrence of 'we' or 'our'-formulations' in the entire manuscript. | Entire manuscript |
| **The title should be made more effective and to reflect better the objectives.** | We agree and improved the title of the manuscript. | Generating a rule-based global gridded tillage dataset |
| **The abstract should summarize better the whole manuscript.** **The Introduction section should be made more fluent and readable.** **The novelty should be explained better and the objectives made clearer.** **Methods should be organized better to allow readers understanding how methods have been used.** **Results and Discussion sections would require to be supported by improved Methods and data section.** | We improved the manuscript in terms of structural adjustments and better separation into sections. Further we improved the abstract, introduction, formulation of objectives, data and method, and the discussion section in the course of this review process as suggested by all referees. | Entire manuscript |
| **Referee #5** | | |
| **The presentation is almost clear, but the English can be improved.** | We carefully checked the language and improved the wording and formulations. | Entire manuscript |
| **Line 58: What is HYDE?** | HYDE stands for 'History Database of the Global Environment'. HYDE is an internally consistent combination of historical population estimates and allocation algorithms with time-dependent weighting maps for land use including grassland but also cropland including its irrigated and rainfed shares. We now explain that abbreviation and have corrected the reference. | Prestele et al. (2018) mapped reported national values of CA area from Kassam et al. (2015) to cropland of the History Database of the Global Environment database (HYDE; Klein Goldewijk et al. (2017)) for the year 2012. |

| | | |
|---|---|---|
| **Line 60: "For downscaling national values Prestele et al. (2018)…" this sentence is too complicated. Should be rephrased.** | We agree and rephrased the sentence. | Based on literature findings, Prestele et al. (2018) developed a CA adoption index per grid cell composed by a set of spatial predictors as aridity, field size, soil erosion, market access, and poverty for downscaling reported national CA area values. Their global map of CA at a spatial grid resolution of 5 arc-minutes is freely available for application in impact assessments in global model simulations. |
| **Line 94: What is ESM?** | Thank you for that hint – we have simply overseen to define this abbreviation. At first occurrence of the word 'Earth system model' in our manuscript we now introduced the abbreviation 'ESM'. | Section 1 |
| **Line 106: I do not understand the sentence "… or can assess different tillage impacts just in form of scenarios". Should be rephrased.** | We agree and rephrased the section. | In the absence of detailed area and tillage type information, the global ecosystem modeling community currently can assess difference of contrasting tillage type impacts just in form of stylized scenarios simulating the effect on the entire cropland area (Del Grosso et al., 2009; Olin et al., 2015; Pugh et al., 2015). One recent exception is the assessment by Hirsch et al. (2018) who assess the effects of an altered albedo from residues used for soil cover on CA areas, using the data of Prestele et al. (2018). |
| **Line 110: "increase understanding of the drivers for different tillage practices". What do the authors mean by "drivers"?** | We agree and revised the section. | The objective of this study is to a) increase the understanding of differences in tillage practices at the global scale b) formulate rules to spatially map tillage systems to the grid scale, and c) develop an open source and open data crop-specific tillage system dataset for the parameterization of tillage events and area in global ecosystem models and assessments. In order to do so we develop a global tillage system classification. Further we analyze underlying causes for the occurrence of different tillage systems and make use of available data in order to map them to a global grid of 5 arc-minutes resolution. |
| **Line 222: "We developed several mapping rules have been in order to allocate the…"this sentence is too complicated. Should be rephrased.** | We agree and improved the sentence. | We developed several mapping rules to allocate the six tillage system to the grid scale, employing a decision tree as shown in Fig. 1. |
| **Line 228: Here the authors mentioned the depth of 15 cm, but claimed that "we decided for a minimum depth of mechanized tillage of 20 cm" above. Please explain this inconsistency (the same for Figure 2).** | We improved figure 1 and the entire calculation for the fraction of rotational tillage crops on soil deeper than 15 but shallower than 20 cm depth to bedrock because of this detected inconsistency. That cropland fraction is now | We applied a downscale algorithm of national reported CA area values on potential CA area (see Fig. 1 box "Downscaling"; see following section for more details). The remaining cropland not being assigned to CA is checked again for soil depth to bedrock. In case it was lower than 20 cm, the cropland was assigned to reduced tillage assuming less depth, frequency, |

| | newly allocated to the reduced tillage system. An updated version of the tillage data set and R-script will be provided in the context of this revision process. | mixing efficiency or alternative cultivation practices. In case of soil depth to bedrock of 20 cm or more the remaining cropland was depending on crop type either mapped to the conventional annual or rotational tillage system following the finding of Kouwenhoven et al. (2002) mentioned above for perennial weed management. |
|---|---|---|
| **Line 533: "global ecosystem models currently run on 0.5° resolution and may have to aggregate the data for input usage" this is not always the case. In many ecosystem models (e.g. ORCHIDEE), their dynamics are simulated at a coarse resolution, but they divide the large model pixel to smaller ones in considering the agricultural processes.** | Thank you for the hint. We see that the sentence was generalizing current spatial resolution in global model simulations too much so we rephrased it also to hint at the uncertainty regarding aggregation. | The resolution of the generated dataset with 5 arc-minutes is quite high. Global ecosystem models are currently mostly run at a coarser resolution than our dataset's resolution and the tillage data may have to be aggregated in such cases. This could introduce further uncertainty to the area under a certain tillage system. |
| **Referee #6** | | |
| **This paper does not adhere to a single unit to describe area. The abstract and discussion use "Mkm^2" while all figures and tables use "ha". Readers cannot compare the different units directly and need to convert "ha" to "Mkm^2". Thus, I recommend that the authors use "km^2" instead of "ha" in all the figures and tables. The main document should be also modified accordingly.** | We agree and harmonized all area unit indication in the text and figures of the entire manuscript to km². | Entire manuscript |
| **I tried to run R script on my PC; however it did not run because it required some data to run in my environment. Thus, I recommend that sample R script should be provided with sample input dataset and output dataset. Then reader can run R script with the data provided and verify their output against a sample output dataset that can be involved. Otherwise, the reader cannot check whether their results were correctly reproduced or not.** | It is correct that in order to run the script the user needs to download the input data sets as indicated in our article, R-code but also described in the meta-data at the repository's websites. We now include sample input and output data which can be applied for testing the functionality of the R-script, when setting 'sample_calc' to 'TRUE' in the beginning of the script. | R-code V.1.1 |
| **In terms of figure 1, I** | Thank you for this | R-code V.1.1 |

| | | |
|---|---|---|
| **suppose that each box in the figure should correspond to the R script. Thus, it is better to add information (ex. line number) that indicates which part of the R script corresponds to the boxes, and also to show the location in the Rscript, where different crop-specific tillage systems are evaluated. According to these, readers can easily understand the R script, which is also the authors' objective.** | suggestion. We have now harmonized the wording of the tillage systems between the manuscript, the data-flow diagram (Figure 1) and the accompanying R-script. We improved the structure of the R-script to be more in line with the steps described in the manuscript and Fig 1. In the R-script we also added comments, indicating which section generates which table and values of the manuscript. | |
| **Own considerations:** | | |
| | Improved Fig. S5, Replaced 12th panel scatterplot with correct output for fields +100% slope increase instead of erosion +100% | Supplement (SI) |
| | Recalculated crop mix and field size interpolation which lead to (small) changes in the entire results so we redid all tables, figures in manuscript and in the online versions of the R-code and data | Entire manuscript, Supplement (SI), R-code V.1.1, and tillage dataset V.1.1 |
| | Improved calculation for reduced and rotational tillage | Entire manuscript, Supplement (SI), R-code V.1.1, and tillage dataset V.1.1 |

[revised manuscript text omitted]
** | **11,314,386** | **100** | **15,714,386** | **100** | **12,160,000** | **100** |

---

## Referee Report (RR1)

This dataset and R script can contribute to our understanding of the generation of a classification of tillage practices, and to map of crop-specific tillage systems for around the year 2005. The paper is acceptable for publication because the authors addressed all review comments. Thus, I recommend that it should be published without any modification.

---

## Referee Report (RR2)

The authors have made a great effort to address all the comments from all reviewers.

I believe the manuscript has substancially improved and now meets the required quality for publication.

Therefore, my recommendation is to publish in its present form.